# ROYAL SOCIETY
# OPEN SCIENCE

statistics/electrical engineering/systems theory

inference, linear stochastic process, mode, Gaussian process, Kalman filter, AC power networks

**Author for correspondence:**
R. S. MacKay
e-mail: R.S.MacKay@warwick.ac.uk

In memory of Professor Sir David John Cameron MacKay FRS (22 April 1967–14 April 2016).

# Inference of dominant modes for linear stochastic processes

## R. S. MacKay[1,2]

[1]Centre for Complexity Science and Mathematics Institute, University of Warwick, Coventry CV4 7AL, UK
[2]Alan Turing Institute, London NW1 2DB, UK

RSM, 0000-0003-4771-3692

For dynamical systems that can be modelled as asymptotically stable linear systems forced by Gaussian noise, this paper develops methods to infer (estimate) their dominant modes from observations in real time. The modes can be real or complex. For a real mode (monotone decay), the goal is to infer its damping rate and mode shape. For a complex mode (oscillatory decay), the goal is to infer its frequency, damping rate and (complex) mode shape. Their amplitudes and correlations are encoded in a mode covariance matrix that is also to be inferred. The work is motivated and illustrated by the problem of detection of oscillations in power flow in AC electrical networks. Suggestions of some other applications are given.

## 1. Introduction

In January 2015, National Grid asked if I could improve their methods for detection of oscillations in power flow, to estimate frequency, damping constant, mode shape and amplitude. Fig. 1 from [1] shows an example where such a mode of oscillation became clear. This type of oscillation is called 'inter-area'; for a review of oscillations in electrical power flow, see [2]. National Grid is interested in detecting such modes in nascent form, so that they can design and install suitable controllers to limit them.

As my brother David, author of [3], was expert in data analysis, I asked him what he would recommend. He responded 'Use a Gaussian process'. It looked a good idea and this paper is the result.

I specialized to linear stochastic processes because they are physically well motivated and the inference (also known as estimation) of their state can be carried out in streaming mode with constant amount of computation per observation. It is a well-developed class, e.g. [4], with major results for inference dating back to the 1960s.

Inference methods for linear stochastic processes are generally used to infer the state of a system with known parameters from observations (the parameters consist of the system matrix, the

**Figure 1.** Voltage angle at seven locations in England relative to CE2 as a function of time. Reproduced with permission from [1]. Angle differences drive power flow, so oscillations in angle differences indicate oscillations in power flow.

noise covariance matrix, the observation function and the observational noise covariance matrix). They can also be used to infer the observable parameters of the system if the others are known or a strong prior probability distribution for them is assumed. This generally requires knowing a significant fraction of the parameters to a significant accuracy in advance. Furthermore, for a large system the remaining parameter space can be high dimensional, making the inference imprecise. If the goal is to infer the possible modes of the system, the transformation from the system matrix to modes (computation of eigenvalues and eigenvectors) can be highly sensitive to the system matrix, adding yet more uncertainty, so it would be better to devise a method to infer the modes directly. Another consideration is that in many circumstances one would want a method that can run in real time, as a monitor and eventually for use in control.

The point of this paper is to present a method to infer the 'dominant' modes, those that have significant amplitude, for a linear stochastic process of many degrees of freedom, with significantly lower dimension of parameter-fit than for the whole system, and to do this in streaming mode. I consider both complex modes (those with oscillatory decay) and real modes (those with monotone decay). What I will propose here has precedents, yet I hope it will be valuable, particularly for AC power flow.

In addition to detecting oscillations in power flow in electricity networks, I envisage the method to be useful in various other contexts, for example detecting soft modes in civil engineering structures, inferring the internal structure of the sun from observation of acoustic waves at the surface (helioseismology), understanding gene expression, and studying business cycles.

The paper starts by presenting my approach to inferring dominant modes. Then it specializes to a method to infer them in streaming mode. An extension is given to filtered Gaussian noise forcing. Next, a formulation of AC power flow network dynamics is proposed, to set it up for potential treatment by the method. Implementation and testing the method on an AC power system, whether real or simulated, is deferred to future work, though some simple data analysis tests are reported here. Finally, a discussion section compares the method with other approaches and proposes some other applications. For the uninitiated, a series of appendices gives a pedagogical introduction to Gaussian processes (GPs), linear stochastic processes and Bayesian inference for them, including Kalman filtering, plus some useful formulae and covariance functions for the AC power model.

## 2. Fitting dominant modes

Suppose an autonomous differentiable dynamical system is subject to random forcing near an asymptotically stable equilibrium $m$. Linearizing about the equilibrium produces a system of the form

$$\dot{x} = A(x - m) + \xi, \tag{2.1}$$

with $A$ an asymptotically stable matrix. Suppose $\xi$ is a (multi-dimensional) Gaussian white noise process with zero mean and auto-correlation $\langle \xi(t)\xi(s)^T \rangle = C^\xi \delta(t-s)$. Modifications of the noise process will be discussed in §4.

Suppose observations are taken at an increasing sequence of times $t_i$ (not necessarily equally spaced), in the form

$$y_i = Z_i x(t_i) + \zeta_i, \tag{2.2}$$

with $Z_i$ being observation matrices (not necessarily all the same) and $\zeta_i$ independent zero-mean Gaussians with covariance $C^{\zeta_i}$, representing measurement error.

From the observations $y$ and knowing $m$, $A$, $C^\xi$, $Z_i$, $C^{\zeta_i}$, a standard approach is to infer a probability distribution for $x(t)$; the method reduces to linear algebra (appendix A.4). The next level of inference is if $m$, $A$, $C^\xi$ are not known but one has a prior probability on their joint distribution, to infer them too from the observations (one could also infer the $C^{\zeta_i}$ if they are not known). This is a nonlinear problem in Bayesian inference but can be tackled by Monte Carlo methods or by gradient methods for the likelihood. After that, one could determine the eigenvalues and eigenvectors of the resulting matrix $A$, thereby inferring the modes of the system, and one could compute their amplitudes and covariance under the noise process with the inferred $C^\xi$.

Instead of the above, I propose to fit the dominant modes of $A$ and their covariance, directly from the observations, without any prior on $A$ or $C^\xi$. The *dominant modes* are those that best explain the observations, in the sense of Bayesian model comparison (appendix A.5).

The idea is that the system matrix $A$ can always be put into a block-diagonal form $D$, i.e.

$$A = BDB^{-1}$$

for some invertible matrix $B$ (that I take real), with the diagonal blocks of $D$ taking simple forms, e.g. a single (negative) real number $-\lambda$ or a $2 \times 2$ block of the form

$$\begin{bmatrix} -\alpha & -\omega \\ \omega & -\alpha \end{bmatrix}.$$

The former case represents a real mode, the latter a complex (or oscillatory) mode. More complicated blocks may be required in the case of multiple eigenvalues. They may also be advisable for groups of nearby eigenvalues, but for present purposes those refinements are ignored. The columns of $B$ are mode shape vectors. Columns for complex modes must be taken in pairs that I call the real and imaginary parts.

Thus, one can think of the state $x$ as being an observation

$$x = m + Bu \tag{2.3}$$

on a process for the amplitudes $u$ of modes:

$$\dot{u} = Du + \eta, \tag{2.4}$$

with $D$ block diagonal, and $\eta$ Gaussian white noise with covariance matrix $C^\eta = B^{-1}C^\xi B^{-T}$. The real observations $y_i$ become observations on the mode process:

$$y_i = Z_i m + Z_i Bu(t_i) + \zeta_i. \tag{2.5}$$

The beauty of this view is that one can then forget about $A$ and $x$ and consider equations (2.4), (2.5) as a self-contained inference problem for $u$, $m$, $D$, $C^\eta$, $B$, given the set of $Z_i$, $t_i$, $y_i$, $C^{\zeta_i}$ (the measurement noise covariance $C^{\zeta_i}$ could also be considered unknown). Furthermore, there is no need to keep $D$, $B$ and $m$ of the same dimension as $A$. If $M$ is the dimension of the space spanned by the rows of all the matrices $K_i$ then one can try to fit equations (2.4), (2.5) to the data for much smaller sizes of $D$ than $A$, say dimension $d \times d$, with $N_r$ and $N_c$ real and complex modes ($d = N_r + 2N_c$), and then fit an equilibrium vector $m$ of dimension $M$ and mode shape matrix $B$ of dimension $M \times d$. The unfitted modes will just contribute to the inferred forcing and measurement noises.

Bayesian model comparison (appendix A.5) allows to compare the evidence for models with different numbers $N_r$ and $N_c$ of real and complex modes. The fit with the highest Bayes factor gives the dominant modes.

Finally, the covariance matrix $S = \langle uu^T \rangle$ for the resulting mode amplitudes is given by (appendix A.3)

$$S = \int_0^\infty e^{Dt} C^\eta e^{D^T t} \, dt, \tag{2.6}$$

which can be evaluated explicitly since $D$ is block diagonal with small blocks. For example, the term corresponding to two real modes $m$, $n$ is $S_{mn} = C^\eta_{mn}/(\lambda_m + \lambda_n)$. Formulae for the other cases can be derived using results in appendix A.7.

There are some redundancies in this specification, which will give rise to problems in maximizing likelihood so need removing. Firstly, the order in which the modes are labelled is irrelevant. One could eliminate this freedom by choosing to list first all the real modes and then all the complex modes and labelling them in order of size of $\lambda$ and $\alpha$, respectively. Secondly, each mode shape vector can be scaled by an arbitrary non-zero scalar (real for a real mode, complex for a complex mode), subject to scaling $C^\eta$ by the inverse square root. Note that since I am using a purely real representation, when I say multiplication of a complex mode shape vector by a complex scalar $x + iy$, I mean to take the linear combinations $xb_r - yb_i$, $xb_i + yb_r$ of the columns $b_r$, $b_i$ of the mode shape vector. One could eliminate this freedom by selecting a 'large' component $i_n$ for each mode $n$ and setting $B_{i_n n} = +1$ for a real mode, $[+1, 0]$ for a complex mode. But as one explores parameter space, one may need to change these choices, so a continuous choice would be preferable.

Also, one needs to enforce $C^\eta$ to be positive semi-definite (PSD). One way to achieve this is to write $C^\eta = e^R$ for $R$ symmetric. There are efficient algorithms for exponentiating matrices. Another is to write $C^\eta = LL^T$ with $L$ lower triangular (in some chosen order on modes), but the diagonal elements of $L$ should be chosen non-negative to remove another redundancy of sign. Such a Cholesky decomposition is a common step for efficient matrix computations so could come for free.

As mentioned above, it might be that a complex mode is close to transition to a pair of real modes, or vice versa. To allow parameter search in a uniform way near such a transition, it would be better to generalize complex modes to also allow pairs of real modes, as in [5], but I leave incorporating that refinement to the future. Similarly, one could allow the formation of non-trivial Jordan blocks and the associated transitions.

# 3. Inferring dominant modes in real time

To infer dominant modes in real time, I take the view expressed in §2, and apply a Kalman filter (appendix A.6). Thus consider a mode process $u$:

$$\dot{u} = Du + \eta, \tag{3.1}$$

with $D$ block diagonal, and observations

$$y_i = Z_i m + Z_i Bu(t_i) + \zeta_i. \tag{3.2}$$

Suppose $Z_i$ and $C^{\zeta_i}$ are known (this would be from calibration and testing of the instruments). It is desired to infer $D$, $B$, $m$ and $C^\eta$. Given all, the Kalman filter enables to infer $u(t)$ in real-time, both its mean and covariance. Then one can calculate a discounted evidence rate for the parameters $D$, $B$, $m$, $C^\eta$, as explained in appendix A.6. One can choose how to seek to maximize this but for definiteness I will describe the Newton method.

Choose numbers $N_R$, $N_C$ of real and complex modes to fit, respectively. Choose a discount rate $\lambda$ for the evidence. Make an initial guess at the mode time-constants: $\lambda_n$ for each real mode, $\alpha_n$, $\omega_n$ for each complex mode, and at the mode shape matrix $B$ (one column for each real mode, two columns for each complex mode), and the mode forcing covariance matrix $C^\eta$. Make an initial guess at the mean vector $m$. All this forms an initial guess for the parameter vector $\mu = (D, B, m, C^\eta)$. Choose initial $u_{0|0}$ and $P_{0|0}$.

When each new observation $y_i$ arrives, note the time $t_i$ it was taken, the observation matrix $Z_i$ and the measurement noise covariance $C^{\zeta_i}$, and set

$$\tau_i = t_i - t_{i-1} \tag{3.3}$$

$$u_{i|i-1} = e^{D\tau_i} u_{i-1|i-1} \tag{3.4}$$

$$G_i = \int_0^{\tau_i} e^{Ds} C^\eta e^{D^T s} \, ds \tag{3.5}$$

$$P_{i|i-1} = e^{D\tau_i} P_{i-1|i-1} e^{D^T \tau_i} + G_i \tag{3.6}$$

$$y_{i|i-1} = Z_i(Bu_{i|i-1} + m) \tag{3.7}$$

$$v_i = y_i - y_{i|i-1} \tag{3.8}$$

$$F_i = Z_i B P_{i|i-1} B^T Z_i^T + C^{\zeta_i} \tag{3.9}$$

$$K_i = P_{i|i-1} B^T Z_i^T F_i^{-1} \tag{3.10}$$

$$u_{i|i} = u_{i|i-1} + K_i v_i \tag{3.11}$$

$$P_{i|i} = (I - K_i Z_i B)P_{i|i-1} \tag{3.12}$$

$$\varepsilon_i = -\frac{1}{2}(v_i^T F_i^{-1} v_i + \log \det F_i + d_i \log 2\pi) \tag{3.13}$$

$$\tilde{L}_i = e^{-\lambda \tau_i}\tilde{L}_{i-1} + \varepsilon_i \tag{3.14}$$

$$\varepsilon_i' = -v_{iT} F_i^{-1} v_i' + \frac{1}{2} v_i^T F_i^{-1} F_i' F_i^{-1} v_i - \frac{1}{2}\text{tr}(F_i' F_i^{-1}) \tag{3.15}$$

$$\tilde{L}_i' = e^{-\lambda \tau_i}\tilde{L}_{i-1}' + \varepsilon_i' \tag{3.16}$$

$$\varepsilon_i'' = -v'^T F^{-1} v' + v^T F^{-1} F' F^{-1} v' + v'^T F^{-1} F' F^{-1} v$$

$$- v^T F^{-1} v'' - v^T F^{-1} F' F^{-1} F' F^{-1} v + \frac{1}{2} v^T F^{-1} F'' F^{-1} v \tag{3.17}$$

$$\tilde{L}_i'' = e^{-\lambda \tau_i}\tilde{L}_{i-1}'' + \varepsilon_i'' \tag{3.18}$$

and

$$\mu_i = \mu_{i-1} - \tilde{L}_i'' \tilde{L}_i' \tag{3.19}$$

The above notation for second derivatives in equations (3.16)–(3.19) is condensed, but the first occurrence of $'$ in each term should be understood as $\partial/\partial \mu_j$ and the second as $\partial/\partial \mu_k$. Also, the subscripts $i$ on all the terms in $\varepsilon_i''$ have been suppressed to save space.

Note that the integral for $G_i$ (in equation (3.5)) is easy to work out because $D$ is block diagonal. Similarly, the update (in equation (3.6)) for $P_{i|i-1}$ is easy. These can be derived from appendix A.7.

The method can also handle the case where the $C^{\zeta_i}$ are not known but are taken from some prior probability distribution. One just adds them to the list of parameters to infer.

If one wants to allow the number of modes to vary then one needs to do Bayesian model comparison, by running several different models alongside each other and computing their Bayes' factors (appendix A.5).

# 4. Filtered noise models

The assumption of forcing by Gaussian white noise might not be realistic in many cases. Perhaps one has to leave the Gaussian world. For example, one can generalize to the world of Student $t$-processes [6] or to elliptic stable processes [7], both of which continue to be specified by a mean function and a covariance kernel. Marginals remain in the same class. Conditioning on $N$ variables increases by $N$ the number of degrees of freedom in a multivariate $t$-distribution. Conditioning stable distributions is not so easy, e.g. [8], but see also [9]. The result of forcing a linear system by a stable process belongs to the same class (but this fails for $t$-processes). I am not aware of an analogue of the Kalman filter to speed up the inference for stable processes in streaming mode, but I leave that for future investigation (see [10]).

On the other hand, there is a class of generalizations of Gaussian white noise that can be incorporated easily in my framework. They are the filtered Gaussian noises, defined as the solution of

$$\dot{\xi} = J\xi + w \tag{4.1}$$

for some asymptotically stable matrix $J$ and $w$ a (multidimensional) Gaussian white noise with covariance $\langle w(t)w^T(s)\rangle = C^w \delta(t-s)$. They fit right in the framework by simply considering the joint process

$$\dot{x} = Ax + \xi \tag{4.2}$$

and

$$\dot{\xi} = J\xi + w, \tag{4.3}$$

which is just a special skew-product form of the general case of a linear system forced by Gaussian white noise $w$. Some component of white noise can also be added to the $\dot{x}$ equation if desired.

Then inference of the dominant modes would also involve inference of the modes of the noise filter $J$, in particular its eigenvalues (its eigenvectors are not observable from the measurements). A curious feature is that I do not see a rational way to assign the eigenvalues between $A$ and $J$.

# 5. AC electricity networks

I turn now to the motivating application.

The dynamics of an AC (alternating current) electricity network can be modelled approximately by a connected graph with a node for each rotating machine (synchronous generator or motor) [11] (this leaves open the question of how to model DC/AC convertors, such as at wind farms, solar photovoltaic farms and DC interconnector terminals). Let $N$ be the number of nodes. As described in [12] (other useful references are [13,14]), one can model an AC network at various levels of complexity. If one ignores aspects like the dynamics of the voltages[1], 3-phase imbalances, reactive power control and harmonics, the state can be specified by a phase $\phi_l$ and frequency[2] $f_l = \dot{\phi}_l$ at each location $l$, and dynamics for the vector $f$ of frequencies and phases $\phi$ are given by balancing power:

$$
\left.
\begin{aligned}
I_l f_l \dot{f}_l &= p_l - \Gamma_l f_l^2 - \sum_{l'} V_l V_{l'} (B_{ll'} \sin(\phi_l - \phi_{l'}) + G_{ll'} \cos(\phi_l - \phi_{l'})) \\
\text{and} \qquad \dot{\phi}_l &= f_l,
\end{aligned}
\right\}
\tag{5.1}
$$

where $I_l$ is an inertia, $\Gamma_l$ a damping constant, $V_l$ is the amplitude of the voltage at $l$, $B_{ll'}$ is a symmetric matrix of ideal admittances of the line between $l$ and $l'$, $G_{ll'}$ is a symmetric PSD matrix of conductances of the line between $l$ and $l'$ (which produces transmission losses) including self-conductances, and $p$ is a vector of power imbalances (generation minus consumption), which is to be regarded as an external stochastic process (e.g. people switching loads on and off, wind farms producing varying power). For the moment, think of $p$ as fixed. For an example of more detailed modelling, see [16].

Note that it is common in the electrical engineering literature (e.g. (1) of [17] or (17) of [18]) to partially linearize equation (5.1) about a reference frequency $f_0$ (usually $100\pi$ or $120\pi$ s$^{-1}$) by writing $\omega_l = f_l - f_0$, $\delta_l = \phi_l - f_0 t$, and replacing $I_l f_l \dot{f}_l$ by $M_l \dot{\omega}_l$ with $M_l = I_l f_0$ (which is often called an inertia again) and $\Gamma_l f_l^2$ by $D\omega_l$ with $D = 2\Gamma_l f_0$. I shall completely linearize later in this section, but for the present retain the fully nonlinear form (equation (5.1)) for discussion of its global phase symmetry and its equilibria.

The system has the special feature of global phase-rotation invariance: if one adds the same constant to all the phases then the dynamics produce the same trajectory but with the constant added. One can quotient by this symmetry group, which we denote by $S$.[3] For example, choose a root node $o$ and a spanning tree in the graph, orient its edges $e$ away from $o$ (other choices are alright but this is to make a definite choice), and let $\Delta_e = \phi_{l'} - \phi_l$ for each edge $e = ll'$ in the spanning tree; there are $N - 1$ of these, and we denote the vector of phase differences by $\Delta$. Then the phase difference between any two nodes can be expressed as a signed sum of the $\Delta_e$, and the equations $\dot{\phi}_l = f_l$ can be replaced by $\dot{\Delta}_e = f_{l'} - f_l$.

The quotient system has a manifold of equilibria in the space of all power imbalance vectors $p$, frequency vectors $f$ and phase difference vectors $\Delta$. For an equilibrium (mod $S$), each node has the same frequency and the phase differences are constant. The manifold of equilibria is a graph of power imbalance vector over the space of common frequency $F \in \mathbb{R}$ (which I take positive) and phase differences $\Delta \in (\mathbb{R}/2\pi\mathbb{Z})^{N-1}$:

$$
p_l = \Gamma_l F^2 + \sum_{l'} V_l V_{l'} (B_{ll'} \sin(\phi_l - \phi_{l'}) + G_{ll'} \cos(\phi_l - \phi_{l'})).
\tag{5.2}
$$

The manifold of equilibria is folded, however, so for given power imbalance vector $p$ there may be 0, 2 or up to $2^{N-1}$ equilibria, of which only some sheets (or parts of sheets) are stable. For equilibria with all phase differences in a suitable subinterval of $(-\pi/2, \pi/2)$, stability can be established by the energy method used in [15], modified to include the conductance matrix $G$ and ignore the voltage dynamics. It should be noted, however, that inclusion of governors or power system stabilizers in the model can destabilize the equilibrium and produce oscillations [12], presumably by a Hopf bifurcation. The method of the present paper is not well adapted to detecting autonomous oscillations as opposed to damped ones forced by noise.

---

[1]This is relatively easy to incorporate, e.g. [15], but a full treatment would require including voltage control, power system stabilizers and excitor control.

[2]As $\phi_l$ is in radians it might be better to denote $f_l$ by $\omega_l$, but I am already using $\omega$ for mode frequencies.

[3]In reality, the system operator is required to keep the phases within some interval (of about 100 cycles) around that for a reference rotor at the nominal frequency, so they exert changes to $p$ to achieve this, thereby breaking the phase rotation invariance, but I ignore that.

Suppose the system is near a stable equilibrium for some $p$. As $p$ moves in time, the response roughly follows it on the manifold of equilibria, but deviations from equilibrium are in general excited and these would relax back to equilibrium if $p$ were to stop moving. For small movements of $p$ about a mean imbalance vector $P$ with corresponding stable equilibrium $(F, \Delta)$, it is appropriate to linearize the system. A reference for small-signal stability in power systems is [19]. Write $\delta f_l$, $\delta\Delta_e$, $\delta p_l$ for the deviations of $f_l$, $\Delta_e$ and $p_l$ from the equilibrium. Write

$$M_l = I_l F, \quad \gamma_l = 2\Gamma_l F \tag{5.3}$$

and

$$T_{ll'} = V_l V_{l'} (B_{ll'} \cos(\Phi_l - \Phi_{l'}) - G_{ll'} \sin(\Phi_l - \Phi_{l'})). \tag{5.4}$$

Then

$$\left.\begin{aligned} M_l \dot{\delta f}_l &= \delta p_l - \gamma_l \delta f_l - \sum_{l'} T_{ll'} (\delta\phi_l - \delta\phi_{l'}) \\ \dot{\delta\Delta}_e &= \delta f_{l'} - \delta f_l \text{ for } e = ll'. \end{aligned}\right\} \tag{5.5}$$

and

Write this as

$$\dot{x} = Ax + C\delta p, \tag{5.6}$$

with

$$x = \begin{bmatrix} \delta f \\ \delta\Delta \end{bmatrix}, \quad C = \begin{bmatrix} \text{diag}M_l^{-1} \\ 0 \end{bmatrix}. \tag{5.7}$$

The power imbalances $\delta p$ are an input to equation (5.6). They fluctuate in time because of variations in generation (in particular, wind and solar) and variations in consumption. I choose to model the dynamics of the power imbalances by

$$\dot{\delta p} = -J\delta p + \sigma\xi \tag{5.8}$$

for some matrix $J$ (with $-J$ asymptotically stable) and (multidimensional) Gaussian white noise $\sigma\xi$ with covariance matrix $K = \sigma\sigma^T$ (later, $J$, $P$, $T$ and $K$ may vary slowly in time). This is a somewhat crude representation, but captures the idea that $p$ has random increments and reversion to a mean. There is evidence that load distribution is close to Gaussian, e.g. fig. 14 of [20], which is consistent with this model, though those data say nothing about the temporal correlations. It is common to neglect temporal correlations of the power imbalance, e.g. [21], but there are automated and human responses to power imbalance which have a filtering effect. One might argue that National Grid's balancing actions are based more on the deviations of the average frequency and phase differences from nominal than the power imbalances, but on the manifold of equilibria these are equivalent.

The resulting system (equations (5.6) and (5.8)) for $(x, \delta p)$ is of the form (2.1). It has a skew-product structure that we could exploit, though that does not play a role for application of my method, so that discussion is deferred to appendix A.8.

So now we can fit observations of $(f, \Delta)$ at as many locations as available (say, $k$) and as a function of time $t$ to a mode processes (2.4) and (2.5) with mean $m$ of the form $(F\mathbf{1}, \bar{\Delta})$ for some $F \in \mathbb{R}$ and $\bar{\Delta} \in \mathbb{R}^{k-1}$, where $\mathbf{1}$ is the vector of length $k$ with all components 1. I make the obvious step of shrinking the spanning tree to one for just the observed nodes. The observations can be deduced from phasor measurement units (PMUs), which measure (among many things) the (voltage) phase relative to a notional 50 Hz reference and the instantaneous frequency at their location.

Let $k$ be the number of PMUs. For $N_R$ real modes and $N_C$ complex modes and

$$M = 2k - 1$$

observation components ($f_l$ for each PMU $l$ and $\Delta_e$ for the voltage phase difference along each edge $e$ in the spanning tree of the PMUs), the parameter space consists of $N_R$ decay rates $\lambda_n$ for the real modes, $N_C$ frequencies $\omega_m$ and decay rates $\alpha_m$ for the complex modes, $N_R$ vectors $B_{in}$ of length $M$ for the real mode shapes normalized to have one component +1, $N_C$ pairs of vectors $B_{im}$ of length $M$ for the complex modes normalized to have one component $(+1, 0)$, $d(d+1)/2$ coefficients of the mode correlation matrix $C^\eta$ (symmetric), where

$$d = N_R + 2N_C,$$

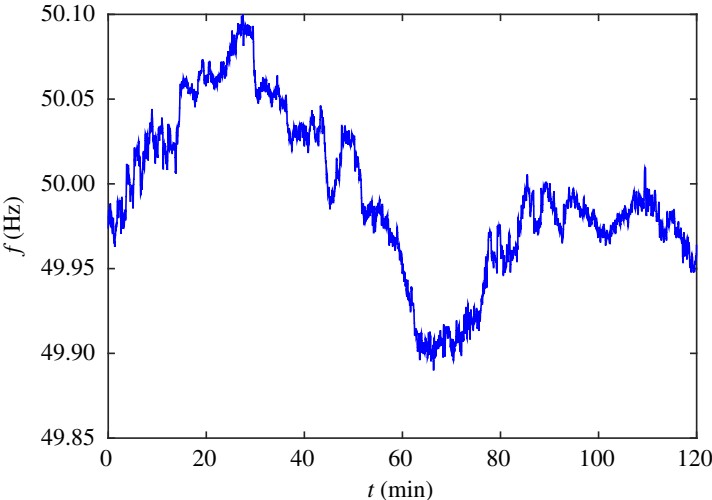

**Figure 2.** A frequency trace over 2 h from National Grid [22].

one mean frequency $F$ and $k-1$ mean phase differences along the edges of the spanning tree. This makes a total dimension

$$d\left(2k + \frac{d-1}{2}\right) + k$$

of parameter space. This is slightly less than the dimension stated in appendix A.4, because for the AC electricity system it is automatic that the time-mean frequencies at all PMUs are the same. If one desires to fit many modes, this dimension could be quite large, but it is still much smaller than the dimension of the parameter space for the whole system.

As an example, if there are $k = 10$ PMUs and one wishes to fit two real modes and one complex mode then $d = 4$ and the parameter space has dimension 53. One might say one is not interested in real modes but some of them are probably the biggest ones and to detect a complex mode accurately one needs to fit the biggest behaviour too.

There is the question of how many modes to allow, both real and complex. This can be decided by the Bayesian comparison method already mentioned (appendix A.5).

One could expect the most important mode behaviour to be an Ornstein–Uhlenbeck (OU) process (see appendix A.2) for $f_o$, assuming $o$ to be a central node for the network. Indeed, using GPML, I found that a 2 h trace of frequency at 1 s intervals, figure 2, which was publicly available from National Grid [22], fit reasonably well to an OU process with a decay time of about 30 min and amplitude 0.045 Hz. The time constant of 30 mins is so long compared to the period (about 2 s) or decay time (about 20 s) of typical inter-area oscillations that it is hardly relevant, and one could just say that on a timescale of up to a minute the basic behaviour of $f_o$ is a Wiener process (continuous-time limit of a random walk) rather than OU. The inferred decay time is a significant fraction of the duration of the time series, so might not be determined very accurately.

On shorter timescales, however, the data look differentiable (figure 3). This is my principal reason for rejecting the hypothesis (e.g. [21]) that power imbalance is a white Gaussian noise, because that would make frequency a nowhere differentiable function of time. Instead, I propose that power imbalance is a first-order filtered white Gaussian noise. Analysis of the power spectrum of fluctuations in the frequency support this proposal. Figure 4 shows a loglog plot of the power spectrum of the data of figure 2 multiplied by a Hann window function ($\sin^2(\pi t/T)$, where $T = 7200$ s is the duration of the series) to prevent the jump between the values at the two ends provoking high-frequency components. The main part of figure 4 has a slope near $-2$, consistent with frequency being an OU process. But for frequency larger than 0.04 Hz (period 25 s) the slope steepens, plausibly to $-4$, until the fact that the data were provided at only 1 s intervals causes an inevitable flattening off of the power spectrum at the Nyquist frequency of 0.5 Hz. National Grid have the data at $1/50$ s intervals, but that is confidential so I cannot use it here. Otherwise, we could see if the slope $-4$ extends to higher frequency. Reference [23] shows a power spectrum in which a slope of roughly $-4$ goes from 0.9 Hz to 3 Hz, but it is for power flow on some line rather than frequency at a node.

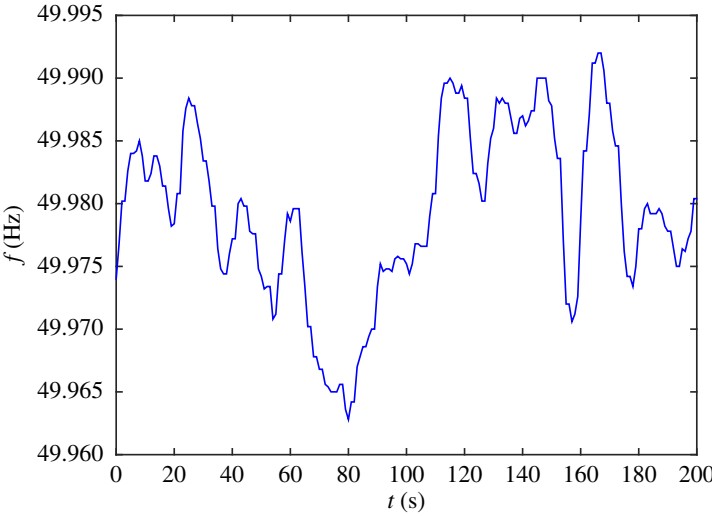

**Figure 3.** The first 3 min 20 s of the frequency trace.

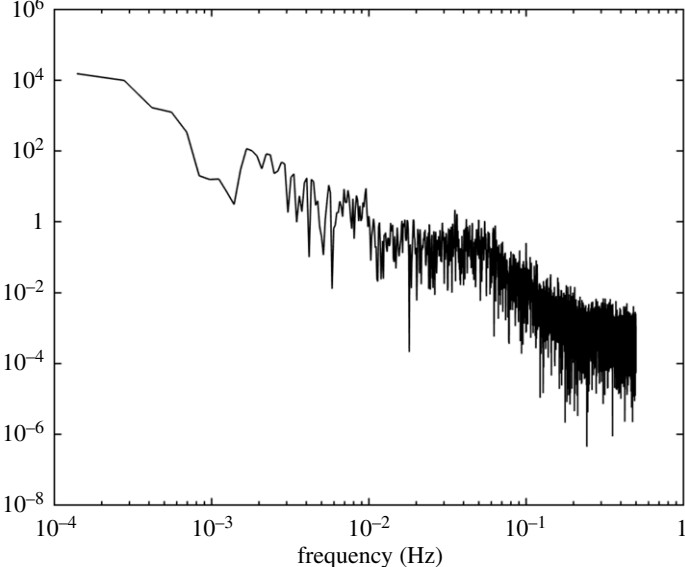

**Figure 4.** Loglog plot of the power spectrum of the data of figure 2 using a Hann window.

A simple model for the data is a first-order filtered OU process (FOU). To justify this, imagine the system is aggregated to a single generator. Then we have two equations of the form

and
$$\left.\begin{array}{l} M\dot{\delta f} = -\gamma \delta f + \delta p \\ \dot{\delta p} = -J\delta p + \sigma \xi. \end{array}\right\} \tag{5.9}$$

It follows from the second equation that $\delta p$ is OU with covariance function $k(\tau) = (\sigma^2/2J)e^{-J|\tau|}$. Then applying equation (A 21) we see that $\delta f$ is a GP with covariance function

$$C(\tau) = \int_0^\infty ds \int_{-\infty}^{\tau+s} d\tau' h(s)k(\tau')h(\tau + s - \tau'), \tag{5.10}$$

where $h$ is the impulse response for the first equation, viz. $h(s) = (1/M)e^{-\Gamma s}$, with $\Gamma = \gamma/M$. Computation of the integral (for the generic case $\Gamma \neq J$) yields

$$C(\tau) = \frac{\sigma^2}{2JM\gamma(\Gamma^2 - J^2)}(\Gamma e^{-J|\tau|} - Je^{-\Gamma|\tau|}). \tag{5.11}$$

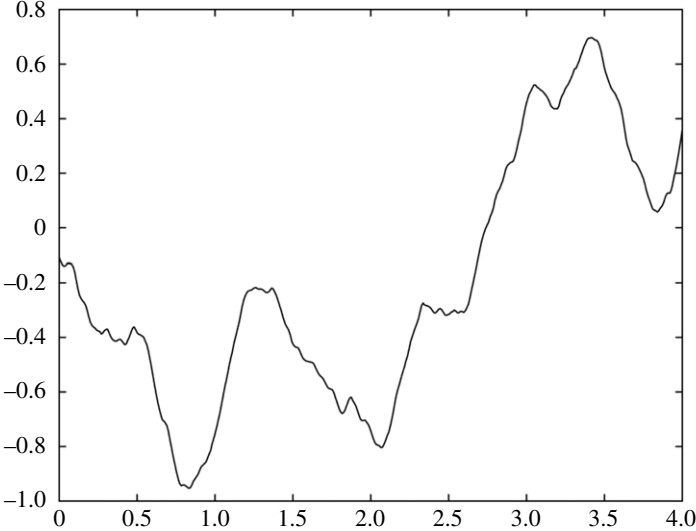

**Figure 5.** A sample from the filtered OU process for $\Gamma = 1/e$, $J = e^2$.

A sample from the FOU process is shown in figure 5. Note that the same covariance function arises for the overdamped linear Langevin process, with $-\Gamma$ and $-J$ being the two real eigenvalues.

Fitting an FOU to the 2 h of data with GPML yields maximum-likelihood estimates for the time constants $1/\Gamma$ and $1/J$ around 11.1 min and 1.87 s, though one can not say from the data analysis which is which. On the basis of estimates of UK system parameters, Andrey Gorbunov (2021, private communication) suggests the most likely match is $1/\Gamma \approx 1.87$ s, and hence $1/J \approx 11.1$ min, making additional evidence that the power imbalance noise is not white on the timescale of interest. It is again awkward that the data are not available at more frequent intervals than 1 s, as the determined time constant 1.87 s is close to this limit. A more thorough treatment would evaluate the posterior uncertainty in the parameter fits and attempt to resolve the discrepancies between the previously estimated OU time constant of 30 min and the current one of 11.1 min, and between the eyeball estimate $f = 0.04$ Hz from figure 4 of where the slope changes, giving a time constant of $(2\pi f)^{-1}$ around 4 s, and the current one of 1.87 s.

It would also be good to apply the approach of §3 to the data to see if two real modes are justified. In particular, one could compare to see whether the model (5.9) is justified, which corresponds to a special subcase of two real-mode fitting in which the mode-noise forcing is perfectly correlated and the mode observation matrix is precisely determined to make only indirect influence of the noise on $f$.

Over long timescales, deviations from Gaussianity have been established [24]. Nevertheless, I believe this does not invalidate Gaussian modelling for short times.

To take this project further, one should next tackle simultaneous readings from two PMUs. This would need the Kalman filter and its parameter-fitting coding up to allow for a number of real and complex modes. It would be best to test it first on simulated data from a power system model with, for example, two generators, one load and a noise process for the power imbalance. Then it could be tested on real data: the phase difference between the two PMUs and their two frequencies.

## 6. Discussion

I have presented a method to detect oscillations in systems with many components. It also detects real modes. It is promising because it can integrate data from many locations simultaneously to enhance the sensitivity of detection of modes of oscillation, and it can run in real-time with constant computation time per observation.

Some references on the problem of calculating modes and mode shapes from phasor measurement units PMU in an AC electrical network are [25,26]. The authors of [27] consider the problem to have been solved. They cite [28–31]. I am not so convinced, because these papers depend to some extent on external estimates of system parameters. It may of course be good to use all available knowledge, but the idea I present here is that one could determine modes of oscillation without determining any system parameters in advance. I think it would be good to try the method of this paper on that

problem, particularly the streaming version. It could also help in determining the time constant for inertia in an AC power system, of crucial importance for its operation.

Detection of modes of oscillation is important in many other contexts. One example is to detect soft (i.e. lightly damped) modes for civil engineering structures such as buildings and bridges, e.g. [32] and ch. 13 of [33]. Another is the identification of modes of oscillation in the sun (helioseismology), which enables to deduce its temperature and rotation profiles[4] [34]. A third is the analysis of gene expression data, e.g. [35]. A fourth is the analysis of business cycles, e.g. ch. 4 of [36], which have been seen for a long time but are still not understood. Use of the Kalman filter to evaluate the likelihood function for parameters is a standard part of training in econometrics, e.g. [37], but I had not seen it used to analyse business cycles until the recent paper [38], which remarkably uses a detection of dominant modes approach, as here.

Detection of oscillations is a very old subject, so I next give a brief review of traditional methods.

A standard approach to detecting oscillations is to identify peaks in the Fourier spectrum [33] or variants [39]. For example, the response $x$ of the second-order system

$$m\ddot{x} + \beta\dot{x} + kx = \eta \tag{6.1}$$

to noise $\eta$ with power spectrum $P$ has power spectrum

$$|\hat{x}(\Omega)|^2 = \frac{P(\Omega)}{(k - m\Omega^2)^2 + \beta^2\Omega^2} \tag{6.2}$$

as a function of frequency $\Omega$. So if the noise is white ($P$ is constant), then the inverse quality factor $Q^{-1} = \beta/\sqrt{mk}$ is precisely the fullwidth at half maximum for the power spectrum $\Omega^2|\hat{x}(\Omega)|^2$ of the velocity $\dot{x}$ (its maximum is at $\Omega_{\text{res}} = \sqrt{k/m}$, known as the resonant frequency), and the damping ratio $\zeta = (1/2)Q^{-1}$ is the halfwidth at half maximum. For $P$ slowly varying on the scale of $\beta/\sqrt{mk}$, the results remain good approximations. This was given a sound grounding in Bayesian analysis (see [40] for a survey and [41] for a pedagogical presentation). It was employed by an MSc student Tajhame Francis on National Grid data, with a view to extending to multidimensional time series. However, it still suffers from issues like dealing with trends, choosing windowing functions, missing data, failure to cater for slowly shifting phase, and poor theoretical justification for taking more than the largest peak if one wants to infer more than one mode of oscillation. Nonetheless, after the initial submission of my paper, [23] was brought to my attention, in which Bayesian spectral analysis (also known as operational modal analysis) is performed with promising results.

Wavelet transforms are popular for resolving signals in both time and frequency (up to the limits of the uncertainty principle), but I am not aware whether they can give an estimate of damping rate.

Another approach is to study the effect of excitation by an impulse (the Prony method and variants like MUSIC and ESPRIT, e.g. [42,43]), but many real-world systems may not be subjectable to impulses. For a review of these and some other methods (e.g. Hilbert transform), see [44].[5]

To detect periodic components in a signal, my brother David [45] proposed the family of stationary GPs with covariance function of the form

$$k(t) = \sigma^2 \exp\left(-\frac{2\sin^2(\omega t/2)}{\lambda^2}\right), \tag{6.3}$$

for which samples are exactly periodic with period $2\pi/\omega$. A slight modification was used in [46] to remove the effect of its non-zero mean, namely

$$k(t) = \sigma^2 \frac{\exp(\lambda^{-2}\cos\omega t) - I_0(\lambda^{-2})}{\exp(\lambda^{-2}) - I_0(\lambda^{-2})}, \tag{6.4}$$

where $I_0$ is a modified Bessel function of the first kind. As $\lambda \to \infty$, it takes the limiting form

$$k(t) = \sigma^2 \cos(\omega t), \tag{6.5}$$

called the Cos kernel, which has the property that it forces anti-periodicity with anti-period $\pi/\omega$: $f(t + \pi/\omega) = -f(t)$. Although these have found valuable uses, and can be made less rigid by multiplication by a decaying kernel such as $\exp(-\alpha|t|)$ (which with the Cos kernel produces OUosc

---

[4]This is achieved with great precision already, but it might be that my approach would have advantages in some circumstances.

[5]Yet another method to determine the eigenvalues of an asymptotically stable system from the response to an impulse is to Laplace transform the response numerically and then fit a Padé approximation and read off its poles.

of [35] or the 'exponentially decaying cosine' of [47]), it seems to me highly preferable to start from the point of view of a linear system forced by noise, which furthermore allows for efficient treatment in streaming mode.

The approaches that are closest in spirit to this paper are 'reduced order' methods such as 'subspace identification' e.g. [48], 'dominant mode analysis' e.g. [49], and 'dynamic mode decomposition' e.g. [50]. The idea of subspace identification is, given observations of some input and output functions in time, to infer a state-space model for the system. The state variables do not have to correspond to any physical variables. My abstract modes are examples. The Kalman filter is used to do the inference. Thus, perhaps my method should be seen as a variant of subspace identification. A difference is that [48] talks a lot about projections, and even the use of the word 'subspace' suggests that they are looking for a subspace of some larger space. I do not have any such larger space and I have no projections. It is possible, however, that if one gets to the bottom of the comparison, the similarities outweigh the differences.

The idea of dominant mode analysis is to subject the system to an excitatory signal, e.g. an impulse or a random binary sequence, and fit a low-dimensional linear model to the sequence of observations as functions of the forcing sequence, allowing for a measurement noise. The differences are that I obtain eigenvectors and use unknown natural forcing.

The idea of dynamic mode decomposition is to infer a linear recurrence relation for a time-series of observations and then to find its eigenvalues and eigenvectors (as time-sequences). There is an implicit assumption of white noise. This is reviewed in [50] along with extensions and relations to some other methods, notably the eigensystem-realization algorithm and linear inverse modelling. It was used by [31] in the power-system context. My eigenvectors are in a space of simultaneous quantities rather than time-delayed, and I do not require equal time intervals between observations.

Next, I discuss deficiencies of my method. One defect is that the forcing might not be Gaussian. For example, even a compound Poisson process with independent Gaussian amplitude is not Gaussian. Also, a consequence of the Gaussian assumption is that the covariance of the response is time-symmetric, as shown in equation (A 23), whereas this might not be true for real systems. As already mentioned, evidence for Gaussian distribution of electrical load is given in fig. 14 of [20], but this reference does not report on time-correlation. Load variations are likely to be the sum of many small independent factors, however, which would make them Gaussian if they have finite variance, by the central limit theorem, but not particularly white. Wind power is far from Gaussian and has long-time dependency: there is considerable research on the statistics of wind power, e.g. [51–53]. Some directions to allow fat-tailed distributions were discussed in §4.

Another defect of the approach of the present paper is that it does not allow for nonlinearity. Nevertheless, for small fluctuations around an equilibrium, linearizing is a good approach. It will fail to give a good approximation, however, if the eigenvalues of any mode approach or cross the imaginary axis. A big question with power-flow oscillations, gene expression and business cycles is whether there is a limit cycle of some underlying deterministic dynamics, or just lightly damped oscillations around an equilibrium forced by noise. Figure 1 suggests that there was a Hopf bifurcation, but the general interpretation of such events in the power system community is that the oscillations are transient, triggered by a switching event, e.g. [1]. For gene expression this dichotomy has been addressed by [54]. For business cycles, most economists decided long ago that they are just a near-unit-root process (meaning lightly damped oscillations forced by shocks) [36], though Grandmont proposed deterministic models with a variety of forms of dynamics [55]. [56] fit a vector autoregressive (VAR) model, but with perhaps more free parameters than justified by the data. Our approach would restrict to a small number of modes, as has been done in the recent paper [38].

A catch with the discounted evidence approach I suggested to allow fitting slowly varying parameters is that the parameters might sometimes vary faster than the chosen memory time-constant. Indeed, this would be a problem for the case for figure 1. It might be better to make a probabilistic model of parameter variation that allows jumps. Large mismatch between prediction and observation in the Kalman filter could be used as an anomaly detector to decide when to insert such jumps.

An interesting issue is that if the noise is considered to be the result of filtering white noise then our method also finds the modes of the filter. Without further information about the structure of the system or direct observations of the forcing process, I see no way of distinguishing between modes of the filter and modes of the system from observations of just the system. An example of this was given in §5.

Lastly, I have not yet tested the method. There will doubtless be issues that arise once one starts to implement it. A likely one is that the choice of normalization condition on the mode vectors and the ordering of the modes by time-constants are both discontinuous and could lead to awkwardness in the fitting; it would be better to find some continuous ways of removing these redundancies. Another

is that the computations may turn out to take more than one-fiftieth of a second and thus not be implementable in real time; then alternating between prediction and correction in the Kalman filter on a slower timescale might be used. A third is that the discounting time for the evidence might turn out to require careful tuning; or as mentioned above, one might do better to switch to a probabilistic model of parameter variation that allows jumps. A fourth is that a particle swarm with many different parameter values and different choices of numbers of modes all running in parallel might be the best way to manage parameter variation; then one will require to implement a good Bayes' factor comparator on top of all the Kalman filters, to decide which particles to mutate and which ones to terminate.

Nevertheless, I think the savings in dimension and the ability to run in streaming mode make my method promising.

Data accessibility. The frequency trace used in §5 was obtained from a publicly available site www2.nationalgrid.com/ Enhanced-Frequency-Response.aspx. That link appears to be no longer active so I have created a Dryad entry containing the excel file at doi:10.5061/dryad.z34tmpgb4 [22].

Competing interests. The author is an associate editor of RSOS but has not been involved in the assessment of the submission.

Funding. The beginning of the work was supported by National Grid under Network Innovation Allowance award NIA_NGET0161. The later parts were supported by the Alan Turing Institute under award no. TU/B/000101.

Acknowledgements. I am grateful to Ben Marshall of National Grid for proposing the problem of detecting inter-area oscillations in January 2015, and to him and his colleague Phillip Ashton for helpful discussions on the topic and pointers to the literature; to MSc student Tajhame Francis for initial investigations by spectral analysis; to my brother David for telling me to 'Use a Gaussian process'; to PhD student Marcos Tello Fraile and postdoc Lisa Flatley for trying to follow my suggestions; to Hannes Nickisch and Colm Connaughton for helping me implement my resulting solutions in GPML; to Carl Rasmussen and Hannes Nickisch for having created GPML; to Zoubin Ghahramani for answering some questions about GPs; to Igor Mezic and Yoshihiko Susuki for discussions on modelling AC networks; to undergraduate summer project student John Prater for coding up the $2 \times 2$ underdamped linear Langevin covariance for GPML; and to Chris Williams, Darren Wilkinson, Keith Worden and especially Janusz Bialek and Andrey Gorbunov, and the reviewers for many useful comments and questions.

# Appendix A

## A.1. Gaussian processes

A GP on a set $T$ is a probability distribution for functions $F : T \rightarrow \mathbb{R}$ such that for all $n \geq 1$ the marginal density $P$ for the vector of values $f_1, \ldots f_n = F(t_1), \ldots F(t_n)$ at any finite sequence $t_1, \ldots t_n \in T$ is Gaussian. Examples for the set $T$ are $\mathbb{R}$ representing time, or the set $V$ of vertices in a graph representing spatial locations in a network, or $\mathbb{R} \times V$ for time and vertices, or $\mathbb{R} \times V \times I$ where $I$ is a set of labels representing components of a vector of values at each vertex and time.

A basic theorem (e.g. [57]) for a GP is that there is a 'mean' function $M : T \rightarrow \mathbb{R}$ and a positive-definite 'covariance' function $C : T \times T \rightarrow \mathbb{R}$ such that

$$P(f_1, \ldots f_n) = (2\pi)^{-n/2} (\det c)^{-1/2} e^{-(1/2)(f-m)^T c^{-1}(f-m)}, \tag{A 1}$$

where $m$ is the vector with components $m_i = M(t_i)$ and $c$ is the matrix with components $c_{ij} = C(t_i, t_j)$. A function $C$ of two variables is said to be positive-definite if for all $n \geq 1$, $t_1, \ldots t_n \in T$ and $v_1, \ldots v_n \in \mathbb{R}$ not all zero then $v^T c v > 0$.

It is convenient to extend the concept of GP to degenerate cases by allowing $C$ to be PSD ($v^T c v \geq 0$). In this case, $c$ may fail to be invertible but the above formula for the density $P$ can be understood as the product of a delta-function on the null space of $c$ and a Gaussian of complementary dimension on the range of $c$, centred at $m$. More formally, its characteristic function $\phi(s) = \langle e^{i s^T f} \rangle = e^{i s^T m - \frac{1}{2} s^T c s}$ for $s \in \mathbb{R}^n$.

Given a GP and observations of a realization of it at a subset $T' \subset T$, possibly with an assumed Gaussian distribution for measurement error (essential if the covariance is not positive-definite), then conditioning on the observations produces a posterior probability distribution for the realization, which is again a GP. It has mean function

$$\tilde{M}(t) = M(t) + C(t, T')[C(T', T') + V]^{-1}(Y - M(T')), \tag{A 2}$$

where $Y$ is the column vector of observations $y_i = f(t_i) + \varepsilon_i$ for $t_i \in T'$, $V$ is the covariance matrix of the measurement error vector $\varepsilon_i$ (assumed zero-mean Gaussian), $C(t, T')$ denotes the row vector of $C(t, t_i)$,

$C(T', T')$ the matrix of $C(t_i, t_j)$, and $M(T')$ the column vector of $M(t_i)$. It has covariance function

$$\tilde{C}(s, t) = C(s, t) - C(s, T')[C(T', T') + V]^{-1}C(T', t), \tag{A 3}$$

where $C(T', t)$ denotes the column vector of $C(t_i, t)$.

Given a family of GPs, labelled by one or more parameters, a prior probability distribution on the parameter space, and observations of a realization, then Bayesian inference gives a posterior probability distribution over the joint space of parameters and realizations. In particular, its marginal on the parameter space gives a posterior probability over the parameter space. In general, this can not be computed explicitly, but search algorithms can find the parameter values maximizing the posterior likelihood. In this way, one can infer the parameters. Computational methods can also give an idea of the posterior uncertainty in the parameters.

There are many introductions to GPs, e.g. [45,46,57–59], and software packages to implement them and infer from them, e.g. GPML and GPy.

Much GP modelling, however, seems to me to be ad hoc. A family of covariance functions is chosen, for example to reflect assumed smoothness class or periodicity, the mean function is often set to zero, and a best fit to the data is obtained. Instead, it is better to use known or assumed structure of the system under study to choose a sensible class of models. This strategy is recognized under the names 'hybrid modelling' or 'latent force models', e.g. [60], or data-based mechanistic modelling [49].

For time-dependent systems, in many contexts, a natural class of models is an asymptotically stable continuous-time linear system forced by Gaussian noise. Furthermore, it is often natural to assume the linear system to be autonomous (some say 'time-invariant') and the noise to be stationary, at least on short time-scales. The idea that filtering white noise produces interesting processes is old, e.g. [61–63].[6]

The noise is not necessarily white. I make the assumption that it is the result of forcing some other autonomous asymptotically stable linear system with white Gaussian noise. The end-result of the assumption on the noise is a skew-product asymptotically stable linear system (consisting of the real system and the noise filter) forced by Gaussian white noise.

Another name for linear stochastic systems is continuous-time VAR processes (see append. B.2.1 of [57]). Classic books on the discrete-time version of such models, including inference for them, are [64,65]. In the latter, they are called dynamic linear models. Linear stochastic process models have been used for inference in many contexts, e.g. [35,66,67].

## A.2. Simple examples of linear stochastic system

The simplest example of asymptotically stable linear system forced by Gaussian noise is the OU process:

$$\frac{\mathrm{d}x}{\mathrm{d}t} = -\mu x + \sigma\xi, \tag{A 4}$$

with $x \in \mathbb{R}, \mu > 0, \sigma > 0$ and $\xi$ unit Gaussian white noise (which can be considered as a degenerate GP on $\mathbb{R}$ with mean $M(t) = 0$ and covariance $C(t, t') = \delta(t - t')$). Then Duhamel's formula yields

$$x(t) = \int_{-\infty}^{t} \mathrm{e}^{-\mu(t-s)}\sigma\xi(s)\,\mathrm{d}s, \tag{A 5}$$

which shows that $x$ is a GP with mean zero and covariance

$$C(t, t') = \langle x(t)x(t') \rangle = \frac{\sigma^2}{2\mu}\mathrm{e}^{-\mu|t-t'|}. \tag{A 6}$$

A sample is shown in figure 6. With probability one, samples are continuous but nowhere differentiable [68].

Next consider the linear Langevin process:

$$m\ddot{x} + \beta\dot{x} + kx = \sigma\xi, \tag{A 7}$$

with $m, \beta, k, \sigma > 0$ (cf. [69]). It follows that $x$ is a GP on $\mathbb{R}$ with mean zero and covariance

$$C(t, t') = \frac{\sigma^2}{2\beta k}\mathrm{e}^{-\alpha|\tau|}\left(\cos\omega\tau + \frac{\alpha}{\omega}\sin\omega|\tau|\right), \tag{A 8}$$

where $\tau = t - t', \alpha = \beta/2m, \omega = (1/m)\sqrt{mk - \beta^2/4}$. This formula is most appropriate for the underdamped

[6]Also, I took a computer music course in Princeton in 1978/1979 in which we made artificial human song by filtering white noise.

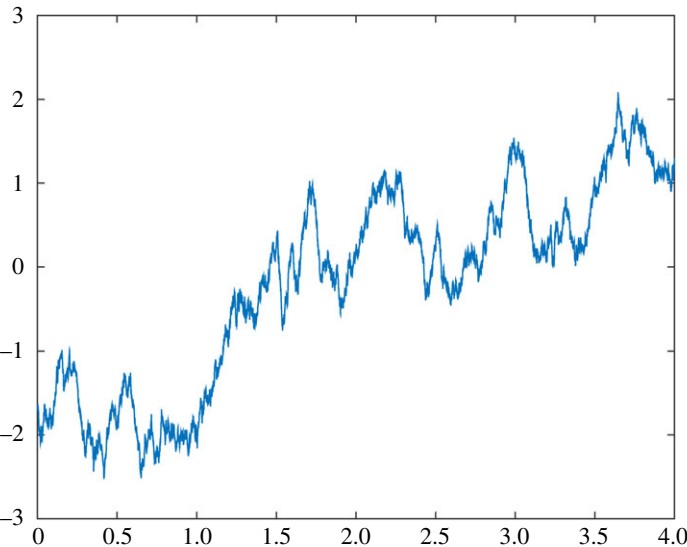

**Figure 6.** A sample from the OU process with $\mu = 1$, $\sigma = \sqrt{2}$.

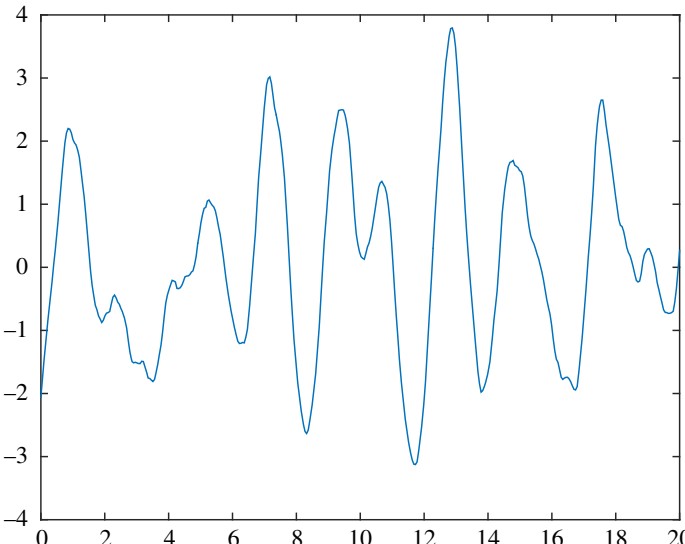

**Figure 7.** A sample for the underdamped linear Langevin process with $\sigma^2 = 2\beta k$, $\alpha = 1/e$, $\omega = e$ ($e$ being the base of natural logarithms).

case $\beta^2/4 < mk$. In the overdamped case $\beta^2/4 > mk$, it is more usefully written as

$$C(t, t') = \frac{\sigma^2}{4\beta k \varepsilon}(\lambda_+ e^{-\lambda_-|\tau|} - \lambda_- e^{-\lambda_+|\tau|}), \tag{A 9}$$

where $\varepsilon = (1/m)\sqrt{\beta^2/4 - mk}$ and $\lambda_\pm = \alpha \pm \varepsilon$. In the critically damped case $\beta^2/4 = mk$, then

$$C(t, t') = \frac{\sigma^2}{2\beta k} e^{-\alpha|\tau|}(1 + \alpha|\tau|). \tag{A 10}$$

Figure 7 shows a sample for an underdamped case, and a sample for an overdamped case appeared in figure 5. With probability one, solutions of the linear Langevin equation are differentiable but nowhere twice differentiable [68].

The linear Langevin equation can be written as a system of two first-order stochastic differential equations. This can be generalized to the two-dimensional system

$$\dot{x} = Ax + \eta, \tag{A 11}$$

where $x \in \mathbb{R}^2$, $A$ a $2 \times 2$ matrix with $\text{tr}\, A < 0$, $\det A > 0$, and $\eta \in \mathbb{R}^2$ with $\eta$ being two-dimensional Gaussian white noise with (PSD) covariance matrix $K$ (i.e. $\langle \eta_i(s)\eta_j(t)\rangle = K_{ij}\delta(t-s)$). Then $x$ is a GP on $\{1, 2\} \times \mathbb{R}$, the first factor indicating the component of $x$ (for which I use subscript notation). It has zero mean. Its covariance function, which I write as a matrix function on $\mathbb{R}^2$ is

$$C(t, t') = \begin{cases} Se^{A^T(t-t')} & \text{for } t > t' \\ e^{A(t'-t)}S & \text{for } t' > t, \end{cases} \tag{A 12}$$

where

$$S = \int_0^\infty d\sigma \, e^{A\sigma}Ke^{A^T\sigma}. \tag{A 13}$$

Taking the first (or second) component of the general two-dimensional system produces a family of covariance functions that we have proposed for purposes such as deciding if a system is under- or over-damped [5].

## A.3. General linear stochastic system

In this section, I review the calculation of the mean and covariance functions for an asymptotically stable continuous-time forced linear system of arbitrary dimension, cf. [57,61,69]. Initially, I allow the system to be non-autonomous and do not restrict the forcing to be Gaussian. Thus consider

$$\dot{x}(t) = A(t)x(t) + \eta(t) \tag{A 14}$$

with $x, \eta \in \mathbb{R}^n$. The asymptotic stability assumption implies that the response $x$ to forcing $\eta$ can be written as

$$x(t) = \int_{-\infty}^t H(t, t')\eta(t')\, dt', \tag{A 15}$$

with $H$ the impulse response (matrix-valued Green function), i.e. the matrix solution of

$$\frac{\partial H}{\partial t} = A(t)H(t, t') \tag{A 16}$$

for $t > t'$ with $H(t' +, t') = I$. Note that for any $t < t' < t''$,

$$H(t, t'') = H(t, t')H(t', t''). \tag{A 17}$$

If $\eta$ is a GP on $\{1, \ldots n\} \times \mathbb{R}$ with mean function $M^\eta$ and covariance function $C^\eta$ (so $C^\eta(s, t) = \langle \eta(s)\eta^T(t)\rangle$), then $x$ is a GP on the same set, with mean function

$$M^x(t) = \int_{-\infty}^t H(t, t')M^\eta(t')\, dt' \tag{A 18}$$

and covariance function

$$C^x(s, t) = \int_{-\infty}^s ds' \int_{-\infty}^t dt' \, H(s, s')C^\eta(s', t')H^T(t, t'). \tag{A 19}$$

If the system is autonomous then $H(s, s')$ is a matrix-function

$$h(\sigma) = e^{A\sigma}$$

of just one variable $\sigma = s - s'$. If the forcing is stationary then $M^\eta$ is constant and $C^\eta(s, t)$ is a matrix-function $k(\tau)$ of $\tau = t - s$ and $k(-\tau) = k(\tau)^T$. So assuming both and changing variables to $\sigma$ and $\tau' = t' - s'$,

$$M^x = \left( \int_0^\infty h(\sigma)\, d\sigma \right)M^\eta = -A^{-1}M^\eta \tag{A 20}$$

and

$$C^x(s, t) = \int_0^\infty d\sigma \int_{-\infty}^{\tau+\sigma} d\tau' \, h(\sigma)k(\tau')h^T(\tau + \sigma - \tau'). \tag{A 21}$$

Now specialize further to white forcing of zero-mean, i.e. $k(\tau) = K\delta(\tau)$ for some PSD symmetric matrix $K$. A common way to write this is $\eta(t) = B\xi(t)$ for $\xi$ a vector of independent unit Gaussian white noises and a matrix $B$; then $K = BB^T$. But $B$ can be replaced by $BO$ for any orthogonal matrix $O$ without changing the probability distribution for $\eta$, so this description contains useless redundancy and it is better to specify the noise $\eta$ by just its covariance matrix $K$.

In this case, $x$ has zero mean and

$$C^x(\tau) = \int_0^\infty d\sigma\, h(\sigma) K h^T(\tau + \sigma) \text{ for } \tau > 0. \tag{A 22}$$

For $\tau < 0$, $C^x(\tau) = C^x(-\tau)^T$. Using $h(\tau + \sigma) = h(\sigma)h(\tau)$ for $\sigma, \tau > 0$ (a special case of equation (A 17)), this boils down to

$$C^x(\tau) = \begin{cases} Sh^T(\tau) & \text{for } \tau > 0 \\ h(-\tau)S & \text{for } \tau < 0, \end{cases} \tag{A 23}$$

where the symmetric matrix

$$S = \int_0^\infty d\sigma\, h(\sigma) K h^T(\sigma), \tag{A 24}$$

giving the covariance of the response of an asymptotically stable autonomous linear system to Gaussian white noise in terms of the impulse response function (cf. (4.5.71a&b) of [61]).

Note that $S$ satisfies a Sylvester equation (actually the special case known as the Lyapunov matrix equation) (cf. (4.5.64) of [61], but with the opposite sign-convention for $A$):

$$AS + SA^T = -K. \tag{A 25}$$

The theory of Sylvester equations (e.g. [70]) shows that this equation has a unique solution for $S$, because $A$ has been assumed to have all its spectrum in the open left half plane, so there are no pairs $(\lambda_i, \lambda_j)$ of eigenvalues for $A$ and $A^T$ that sum to zero. An interesting approach using equation (A 25) to infer $A$ and $K$ from $S$ in the AC electricity context is presented in [21], where the model is called a vector OU process.

## A.4. Bayesian inference for linear stochastic systems

I review the standard Bayesian approach to inference of a linear stochastic dynamical system from observations. The use of the Kalman filter to speed up the computations and to update the inference in real time will be treated in appendix A.6. There are alternative approaches, such as subspace identification methods, e.g. [48], but my goal is not to go into detail, rather to present enough to contrast the approach that I propose in §2.

The system is modelled by

$$\dot{x} = Ax + \xi, \tag{A 26}$$

with $\langle \xi(s)\xi^T(t) \rangle = K\delta(t - s)$. The matrices $A$ and $K$ are considered unknown, though one may have strong prior probability distributions for them. This needs augmenting by a model for the observations, e.g. vectors

$$y_i = Z_i x(t_i) + \zeta_i \tag{A 27}$$

for some known times $t_i$, known observation matrices $Z_i$, and unknown measurement errors $\zeta_i$ that I assume to be independent zero-mean Gaussian vectors with known covariance matrices $H_i$. The idea is that the matrices $Z_i$ specify which components (or combinations of components) of $x$ are measured.

Then the parameters of the model are the matrix elements of $A$ and $K$. If $x$ has dimension $L$, the parameters form a continuous space $\mathcal{P}$ of dimension $L^2 + L(L + 1)/2 = L(3L + 1)/2$ (though this may be reduced significantly if the system has known structure). Denote the parameters compactly by a vector $\mu$.

Previous knowledge about the system is encoded into a prior probability density $P_-(\mu)$ for the parameters. Given the observations $Y = (y_i)$, a posterior probability density $P_+$ is computed for the parameters by Bayes' rule:

$$P_+(\mu|Y) = P_m(Y|\mu)P_-(\mu)/Z(Y), \tag{A 28}$$

where $P_m$ is the probability density for the observations given the parameters, specified by the model, and $Z = \int P_m(Y|\mu)P_-(\mu)\,\mathrm{d}\mu$ is a normalization factor.

If enough observations have been taken (depending on how tight the prior $P_-$ was), then $P_+$ will be tightly peaked around some value $\hat{\mu}$ of the parameter vector. The maximum posterior likelihood value is the $\hat{\mu}$ that maximizes $P_+(\mu\,|\,Y)$ (assuming it is unique). Although the functional form for $P_+$ is not in general computable, numerical algorithms like conjugate gradient ascent can search for $\hat{\mu}$, based on evaluation of $P_+$ (or in practice log $P_+$) and its gradient with respect to $\mu$ at a suitable sequence of points. They can also compute a quadratic approximation to $P_+$ around $\hat{\mu}$ to give an idea of the posterior uncertainty in the inference. Another nice method, called Bayesian optimization, is to fit a GP to the posterior likelihood and optimally choose a sequence of evaluation points to reduce the uncertainty of its maximum [71].

It might be that one is not interested in all the parameters. For example, one might want $A$ but not $K$ (and if one were to treat the measurement noise covariances $H_i$ as unknown they would also appear in the parameter list but probably not be of primary interest). In this case, one could marginalize the posterior probability distribution over the uninteresting parameters, for example compute $P_+(A|Y) = \int P_+(A, K|Y)\,\mathrm{d}K$. In general, this is not possible analytically, but it can be done approximately by various numerical methods. One of the nicest is Bayesian quadrature [72], which fits a GP to the integrand and hence obtains a (one-dimensional) Gaussian distribution for the integral.

Once a best fit to $A$ has been obtained, one could compute the modes of $A$ (frequency, damping, shape) by diagonalization of $A$. If a fit to $K$ has also been obtained, one can calculate the covariance of the response from equations (A 23), (A 24), and hence the covariance of the modes using the diagonalization.

The main point of this paper, however, is that the above approach to inferring modes is overkill. If one is interested only in the dominant modes one can infer them without inferring $A$ and $K$. I say the *dominant modes* are the subset of modes which best explain the observations, in the sense of Bayesian model comparison (highest Bayes' factor, described in appendix A.5). One can always fit the data better by adding more modes but at the expense of making the model bigger and potentially reducing its explanatory power. To infer $d$ modes (counting complex ones twice) from $M$ observation components requires a parameter space of dimension only $(d+1)(M+d/2)$. This is likely to be much less than the dimension $L(3L+1)/2$ of the space of $A$ and $K$ above, because both $d$ and $M$ are smaller than $L$.

The second main point of the paper is that the inference of dominant modes can be run in streaming mode. The description given so far involves collecting all the data and then maximizing the posterior likelihood. This is called *batch mode* of inference. For inference of a process running in real-time, it is better to update the maximum posterior likelihood estimates after each new observation arrives. This is called *streaming mode* of inference. It can be done efficiently, with each new observation requiring the same time to process regardless of how many previous observations have been made, whereas for a general GP the computation time to infer from $n$ observations scales like $n^3$ and the time to take into account one new observation scales like $n^2$ [57].

## A.5. Bayesian model comparison

The number of modes to attempt to fit can be decided by Bayesian model comparison [3]. This is an extension of maximum posterior likelihood search to a setting with two or more models $M_j$, which each have their own continuous parameter spaces $\mathcal{P}_j$. For each model $M_j$, one can compute the posterior probability density $P_+(\mu\,|\,Y, M_j)$ for $\mu \in \mathcal{P}_j$. By various methods, e.g. [73] or Bayesian quadrature [72], one can also compute the normalization constant $Z(Y\,|\,M_j)$, called *Bayes' factor* for the model. Then given prior probabilities $P_-(M_j)$ for the models (which can be taken the same if one is agnostic about which model is best), one applies Bayes' rule again to obtain posterior probabilities

$$P_+(M_j|Y) = Z(Y|M_j)P_-(M_j)/Z(Y), \tag{A 29}$$

where $Z(Y)$ is a normalization factor again, depending on $Y$ and the chosen set of models, but is not required for what follows. This formula can be used to decide which model is the best explanation of the observations and to keep track of near-competitors. For each model $M_j$, the method of appendix A.4. determines best-fit parameters $\hat{\mu}_j \in \mathcal{P}_j$. In our case, the different models correspond to the numbers $N_R$ and $N_C$ of real and complex modes to fit, respectively. The idea is that even though a better fit is achievable with more modes, the required increased dimension of parameter space might not justify it (Occam's razor).

## A.6. Inference from streaming data: Kalman filtering

In many circumstances, it would be preferable to run the inference of modes in real time rather than batch, and efficiently. There are papers on real-time inference with GPs, e.g. [58,66,67,74]. The most important insight is that for linear stochastic processes one can use the Kalman filter, a method that uses the Markovian structure to reduce the computational burden of inference to a constant amount per observation. It goes back to the 1960s and is in many textbooks, though rarely in continuous time. A nice text on recursive estimation more generally is [75].

Here, I review the use of the Kalman filter to infer the state and parameters of a continuous-time linear system forced by white noise from real-time observations. Denote the state of the system at time $t \in \mathbb{R}$ by $x(t) \in \mathbb{R}^n$ and suppose it evolves according to

$$\dot{x} = Ax + \eta, \tag{A 30}$$

with $\eta \in \mathbb{R}^n$ Gaussian white noise of covariance matrix $C^\eta$. Suppose observations are taken at an increasing sequence of times $t_i$. In contrast to claims in some of the literature, these do not need to be equally spaced and one can observe different components of $x$ at different times. Furthermore, one can also allow $A$ and $C^\eta$ to be time-dependent, though for simplicity of exposition I will not do that here.

So let the observations be

$$y_i = Z_i x_i + \zeta_i, \tag{A 31}$$

where $y_i \in \mathbb{R}^{d_i}$, $x_i = x(t_i)$, $\zeta_i \in \mathbb{R}^n$, $Z_i$ are matrices specifying which combinations of components of $x_i$ are observed, and $\zeta_i$ is a zero-mean Gaussian measurement noise with covariance $C^{\zeta_i}$ which I suppose independent for different $i$.

Then for a sequence of vectors $x_i$ at the times $t_i$, use the notation $x_{i|i-1} = \langle x_i | y_{i-1}, \dots y_1 \rangle$ and $x_{i|i} = \langle x_i | y_i, \dots y_1 \rangle$. Similarly, define $y_{i|i-1} = \langle y_i | y_{i-1}, \dots y_1 \rangle$. Let

$$P_{i|i-1} = \langle (x_i - x_{i|i-1})(x_i - x_{i|i-1})^T \rangle \tag{A 32}$$

and similarly $P_{i|i} = \langle (x_i - x_{i|i})(x_i - x_{i|i})^T \rangle$. Write $\tau_i = t_i - t_{i-1}$. As a consequence of the Duhamel formula

$$x_i = e^{A\tau_i} x_{i-1} + \int_{t_{i-1}}^{t_i} e^{A(t_i - t)} \eta(t) \, dt, \tag{A 33}$$

one obtains

$$x_{i|i-1} = e^{A\tau_i} x_{i-1|i-1} \tag{A 34}$$

and

$$P_{i|i-1} = e^{A\tau_i} P_{i-1|i-1} e^{A^T \tau_i} + G_i, \tag{A 35}$$

with

$$G_i = \int_{t_{i-1}}^{t_i} e^{A(t_i - t)} C^\eta e^{A^T(t_i - t)} \, dt. \tag{A 36}$$

Also, averaging equation (A 31),

$$y_{i|i-1} = Z_i x_{i|i-1} + m_i. \tag{A 37}$$

Let

$$v_i = y_i - y_{i|i-1} \tag{A 38}$$

and

$$F_i = \langle v_i v_i^T | y_{i-1}, \dots y_1 \rangle. \tag{A 39}$$

Then

$$F_i = Z_i P_{i|i-1} Z_i^T + C^{\zeta_i}. \tag{A 40}$$

Finally, by conditioning on $y_i$, one obtains

$$x_{i|i} = x_{i|i-1} + K_i v_i \tag{A 41}$$

and

$$P_{i|i} = (I - K_i Z_i) P_{i|i-1}, \tag{A 42}$$

where the 'Kalman gain matrix'

$$K_i = P_{i|i-1} Z_i^T F_i^{-1}. \tag{A 43}$$

The standard use of these equations is to provide an estimate $x_{i|i}$ of the state $x_i$ and its uncertainty (from $P_{i|i}$). But they can also be used to provide the likelihood for the parameters $\mu = (A, C^\xi, m, C^{\zeta_i})$ of the model, given the observations, and this is my primary goal.

To see how to do this, the likelihoods for the observations given the parameter values $\mu$ satisfy

$$P(y_i, \ldots y_1 | \mu) = P(y_i | y_{i-1}, \ldots y_1, \mu) P(y_{i-1}, \ldots y_1 | \mu). \tag{A 44}$$

Now $y_i | (y_{i-1}, \ldots y_1, \mu)$ is Gaussian with mean $y_{i|i-1}$ and covariance $F_i$. So the *evidence* $L_i(\mu)$ for the parameter value $\mu$, defined to be the log-likelihood of the observations as a function of the parameters, updates by

$$L_i(\mu) = \log P(y_i, \ldots y_1 | \mu) = L_{i-1}(\mu) + \varepsilon_i(\mu), \tag{A 45}$$

where

$$\varepsilon_i(\mu) = -\frac{1}{2}(v_i^T F_i^{-1} v_i + \log \det F_i + d_i \log 2\pi). \tag{A 46}$$

Recall that $d_i$ is the dimension of the observation vector $y_i$ at time $t_i$. This provides the total evidence for the given parameters $\mu$, starting from the initial time. Despite the fact that for a general GP it takes time $O(N^3)$ to compute the likelihood from $N$ observations, the class of linear stochastic processes with the above algorithm takes equal time per observation, allowing the computation to be done in real-time.

One can similarly work out how to update the derivative of $L_i$ with respect to the parameters; use

$$\varepsilon_i' = -v_i^T F_i^{-1} v_i' + \frac{1}{2} v_i^T F_i^{-1} F_i' F_i^{-1} v_i - \frac{1}{2} \text{tr}(F_i' F_i^{-1}), \tag{A 47}$$

where prime denotes derivative with respect to any parameter, and propagate the derivatives through the Kalman filter equations. Thus one can make gradient steps to improve the estimate of the maximum-likelihood parameters.

Note that as usual in numerical optimization using gradients, it is best to use an adapted inner product $\mu \cdot v = \mu^T G v$ with $G$ a positive-definite symmetric matrix to compute the gradient $\nabla L$ for the vector $L'$ of derivatives, by $\nabla L = G^{-1} L'$. The inner product should be chosen to reflect the different typical sizes of different components of $\mu$. Ideally, it would be close to the negative of the second derivative of the evidence at the maximum. One has also to decide how far to step along the gradient. There are various choices which go under the name of conjugate gradient methods and also modify the inner product as one goes along (this includes the Davidon-Fletcher-Powell and Broyden-Fletcher-Goldfarb-Shanno methods).

To adapt to the case where the parameters may in reality be time-varying, one could make a GP model for the parameters as functions of time, for example an OU process or slowly varying plus jumps, and infer them. An alternative that I propose as probably more practical is to maximize an exponentially weighted sum of the gains in evidence, allowing one to forget past evidence because it is likely to become irrelevant. Choose a rate constant $\lambda$ for forgetting past evidence. The evidence gained at time $t_i$ relative to $t_{i-1}$ is $\varepsilon_i$ of equation (A 46). An appropriate notion of the weighted sum of gains, that I call *discounted evidence rate*, is

$$\tilde{L}_i = \sum_{j=1}^{i} e^{-\lambda(t_i - t_j)} \varepsilon_j, \tag{A 48}$$

and it updates by

$$\tilde{L}_i = e^{-\lambda \tau_i} \tilde{L}_{i-1} + \varepsilon_i. \tag{A 49}$$

The use of first-order filters such as equation (A 49) to take an exponentially weighted time average, allowing one to forget the distant past, is old. In the Kalman filter context, I have found it used by [76], for example, under the name 'adaptive filtering'. In particular, the idea to first-order filter the evidence gains occurs there, as an anomaly detector. Similarly, it comes in [77]. The approach is analogous to 'fading memory' filters, e.g. [78], but which are for state rather than parameters. Again,

derivative information can be updated and gradient steps made to track maximum-likelihood parameters.

In principle, one can begin by specifying a prior probability density on the parameter space but its effect on the discounted evidence rate will go to zero exponentially in the time since the start, so it may not be very important.

There are issues with doing this optimization in real time. To compute $\tilde{L}_i$ at a new parameter point requires in principle to recompute all the previous $\varepsilon_j$, $j < i$, at the new point. In practice, it is probably enough to compute the new $\varepsilon_j$, $j \geq i$, at the new point and let the earlier ones get forgotten by the discounting. This would slow down each step of the optimization, however, to the discounting rate. Alternatively, one could run many filters in parallel for different parameter values, and use Bayes' rule to combine them. A practical way to deal with pruning and merging the set of such filters is given in [79].

It may be better to also compute the second derivative matrix $\tilde{L}''$ with respect to $\mu$. This can be done by differentiating $\varepsilon'$ again. It gives the ideal choice of parameter step $\Delta\mu$ when near the maximum, namely the Newton step $\Delta\mu = -\tilde{L}''^{-1}\tilde{L}'$. It is not necessary to update $\tilde{L}''$ frequently, an approximation suffices. Thus one could update $\tilde{L}'$ (or at least $\varepsilon'$) as each observation comes in, and $\tilde{L}''$ less frequently.

Note that with the Newton method, it is not in fact necessary to compute $\tilde{L}$ itself. Nevertheless, the ingredients $(v_i, F_i)$ that go into computation of $\tilde{L}$ are required for computation of $\tilde{L}'$, so one might as well track $\tilde{L}$ too.

There may be better optimization methods for this problem. For example, as already discussed, one could compute $\tilde{L}$ at a set of parameter values simultaneously and replace parameter values with low $\tilde{L}$ by ones chosen to be likely to have high $\tilde{L}$ from time to time. Indeed, there are Bayesian optimization procedures that fit a GP to the evidence function and automatically choose the next evaluation point to minimize the variance of the maximum of the posterior GP. Gradient information can be incorporated too.

An alternative to linear stochastic processes and Kalman filtering might be to use a more general stationary GP with equal observation time intervals and fast methods for Cholesky decomposition of the resulting block Toeplitz covariance matrices, e.g. [80–83].

## A.7. Terms involving $e^{Dt}$

For block-diagonal $D$ with blocks of the forms $-\lambda_m$ and $\begin{bmatrix} -\alpha_n & -\omega_n \\ \omega_n & -\alpha_n \end{bmatrix}$, then $e^{Dt}$ is block-diagonal with blocks $e^{-\lambda_m t}$ and $e^{-\alpha_n t}\begin{bmatrix} \cos\omega_n t & -\sin\omega_n t \\ \sin\omega_n t & \cos\omega_n t \end{bmatrix}$.

So for any matrix $C$, $S = e^{Dt}Ce^{D^T t}$ is given by blocks of the following forms:

(i) two real modes: $1 \times 1$ blocks

$$S_{mm'} = e^{-(\lambda_m+\lambda_{m'})t}C_{mm'}$$

(ii) one real mode $m$ and one complex mode $n$: $1 \times 2$ blocks

$$S_{mn} = e^{-(\lambda_m+\alpha_n)t}C_{mn}\begin{bmatrix} \cos\omega_n t & \sin\omega_n t \\ -\sin\omega_n t & \cos\omega_n t \end{bmatrix}$$

(iii) one complex mode $n$ and one real mode $m$: $2 \times 1$ blocks

$$S_{mn} = e^{-(\lambda_m+\alpha_n)t}\begin{bmatrix} \cos\omega_n t & -\sin\omega_n t \\ \sin\omega_n t & \cos\omega_n t \end{bmatrix}C_{nm}$$

(iv) two complex modes: $2 \times 2$ blocks

$$S_{nn'} = e^{-(\alpha_n+\alpha_{n'})t}\begin{bmatrix} \cos\omega_n t & -\sin\omega_n t \\ \sin\omega_n t & \cos\omega_n t \end{bmatrix}C_{nn'}\begin{bmatrix} \cos\omega_{n'} t & \sin\omega_{n'} t \\ -\sin\omega_{n'} t & \cos\omega_{n'} t \end{bmatrix}.$$

In particular $G_i$ of equation (3.5) has blocks of the corresponding types (dropping the suffix $i$ and the superscript $\eta$ from $C^\eta$):

(i)
$$G_{mm'} = \frac{1 - e^{-(\lambda_m + \lambda_{m'})t}}{\lambda_m + \lambda_{m'}} C_{mm'}$$

(ii)
$$G_{mn} = \frac{C_{mn}}{(\lambda_m + \alpha_n)^2 + \omega_n^2} \begin{bmatrix} \mathcal{A} & -\mathcal{B} \\ \mathcal{B} & \mathcal{A} \end{bmatrix},$$

where

$$\mathcal{A} = \lambda_m + \alpha_n - e^{-(\lambda_m + \alpha_n)\tau}((\lambda_m + \alpha_n)\cos \omega_n \tau - \omega_n \sin \omega_n \tau)$$

and

$$\mathcal{B} = -\omega_n + e^{-(\lambda_m + \alpha_n)\tau}(\omega_n \cos \omega_n \tau + (\lambda_m + \alpha_n)\sin \omega_n \tau)$$

(iii)
$$G_{nm} = \begin{bmatrix} \mathcal{A} & \mathcal{B} \\ -\mathcal{B} & \mathcal{A} \end{bmatrix} \frac{C_{nm}}{(\lambda_m + \alpha_n)^2 + \omega_n^2}$$

(ii)
$$G_{nn'} = \begin{bmatrix} ccA - csB - scC + ssD & csA + ccB - ssC - scD \\ scA - ssB + ccC - csD & ssA + scB + csC + ccD \end{bmatrix},$$

where $C_{nn'} = \begin{bmatrix} A & B \\ C & D \end{bmatrix}$, and $cc, cs, sc, ss$ are defined by

$$2cc = \frac{a}{a^2 + \delta^2} + \frac{a}{a^2 + \sigma^2} + e^{-a\tau}\left(\frac{-a\cos\delta\tau + \delta\sin\delta\tau}{a^2 + \delta^2} - \frac{a\cos\sigma\tau - \sigma\sin\sigma\tau}{a^2 + \sigma^2}\right)$$

$$2cs = -\frac{\delta}{a^2 + \delta^2} + \frac{a}{a^2 + \sigma^2} + e^{-a\tau}\left(\frac{\delta\cos\delta\tau + a\sin\delta\tau}{a^2 + \delta^2} - \frac{\sigma\cos\sigma\tau + a\sin\sigma\tau}{a^2 + \sigma^2}\right)$$

$$2sc = \frac{\delta}{a^2 + \delta^2} + \frac{a}{a^2 + \sigma^2} + e^{-a\tau}\left(\frac{-\delta\cos\delta\tau - a\sin\delta\tau}{a^2 + \delta^2} - \frac{\sigma\cos\sigma\tau + a\sin\sigma\tau}{a^2 + \sigma^2}\right)$$

$$2ss = \frac{a}{a^2 + \delta^2} - \frac{a}{a^2 + \sigma^2} + e^{-a\tau}\left(\frac{-a\cos\delta\tau + \delta\sin\delta\tau}{a^2 + \delta^2} + \frac{a\cos\sigma\tau - \sigma\sin\sigma\tau}{a^2 + \sigma^2}\right)$$

with $a = \alpha_n + \alpha_{n'}$, $\sigma = \omega_n + \omega_{n'}$, $\delta = \omega_n - \omega_{n'}$.

The covariance matrix $S$ of equation (2.6) is the case $\tau = +\infty$ of $G$.

## A.8. Skew-product structure of the power system model

The power system model (5.6) and (5.8) has a skew-product structure, namely $\dot{\delta p}$ does not depend on $x$ (also the $x$-dynamics has structure in that it is only the frequency deviations $\delta f$ that see $\delta p$ directly). In reality, perhaps $\dot{\delta p}$ does depend a little on $x$, e.g. National Grid balancing operations and frequency-sensitive generators and loads, but I continue with this model. One way to exploit the skew-product structure is to derive the covariance function for $\delta p$ using equation (A 23) and then insert this into the formula (A 21) for the covariance function of $x$, but it leads to an integration whose treatment is not simple. Alternatively, one can apply equation (A 23) to the joint systems (5.6) and (5.8), exploit the skew-product form of the impulse response, and take the $xx$-block of the covariance function. I chose the latter approach, subject to the simplifying but generic assumption of simple eigenvalues for the full system.

The impulse response of equation (5.8) can be written in matrix exponential notation as $\delta p(t) = e^{-Jt}$. Similarly, the impulse response of equation (5.6) can be written as $x(t) = e^{At}$. To compute the response of $x$ to an impulse on $\dot{p}$, it is convenient to assume that $A$ and $-J$ have no eigenvalues in common, as is generically the case. Then there exists a unique solution $E$ to another Sylvester equation

$$AE + EJ = C, \tag{A 50}$$

and defining $y = x + Ep$ shows that $\dot{y} = Ay + E\xi$. So the response of $y$ to an impulse on $\dot{p}$ is $e^{At}E$. It follows that the response of $x = y - Ep$ to an impulse on $\dot{p}$ is

$$x(t) = h_{xp}(t) := e^{At}E - Ee^{-Jt}. \tag{A 51}$$

Note that using equation (A 50), the time-derivative of $h_{xp}$ at $t = 0$ is just $C$. Thus the impulse response of the full system has the block form

$$h(t) = \begin{bmatrix} h_{pp} & h_{px} \\ h_{xp} & h_{xx} \end{bmatrix}(t) = \begin{bmatrix} \mathrm{e}^{-Jt} & 0 \\ \mathrm{e}^{At}E - E\mathrm{e}^{-Jt} & \mathrm{e}^{At} \end{bmatrix}. \tag{A 52}$$

Then the stationary covariance matrix $S$ (A 24) of the joint process has block form

$$\begin{aligned} S &= \int_0^\infty h(\sigma) \begin{bmatrix} K & 0 \\ 0 & 0 \end{bmatrix} h^T(\sigma)\, \mathrm{d}\sigma \\ &= \int_0^\infty \begin{bmatrix} \mathrm{e}^{-J\sigma}K\mathrm{e}^{-J^T\sigma} & \mathrm{e}^{-J\sigma}Kh_{xp}^T(\sigma) \\ h_{xp}(\sigma)K\mathrm{e}^{-J^T\sigma} & h_{xp}(\sigma)Kh_{xp}^T(\sigma) \end{bmatrix} \mathrm{d}\sigma. \end{aligned} \tag{A 53}$$

It follows from equation (A 23) that (for $\tau > 0$)

$$\begin{aligned} C^x(\tau) &= S_{xp}h_{xp}^T(\tau) + S_{xx}\mathrm{e}^{A^T\tau} \\ &= \left( \int_0^\infty h_{xp}(\sigma)K\mathrm{e}^T\mathrm{e}^{A^T\sigma}d\sigma \right)\mathrm{e}^{A^T\tau} - \left( \int_0^\infty h_{xp}(\sigma)K\mathrm{e}^{-J^T\sigma}d\sigma \right)\mathrm{e}^{-J^T\tau}\mathrm{e}^T. \end{aligned} \tag{A 54}$$

Thus the covariance of $x = (\delta f, \delta\Delta)$ is a linear combination of functions from the impulse response of $x$ to $\dot{x}$ and of $p$ to $\dot{p}$.[7]

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
