## [Peer Review File · Royal Society Open Science]

Review History

RSOS-201442.R0 (Original submission)

Review form: Reviewer 1

Is the manuscript scientifically sound in its present form?

No

Are the interpretations and conclusions justified by the results?

Yes

Is the language acceptable?

Yes

Do you have any ethical concerns with this paper?

No

Have you any concerns about statistical analyses in this paper?

No

Recommendation?

Reject

Comments to the Author(s)

This paper addresses the development of methods to infer dynamical systems dominant modes from observations in real time. Overall, the contribution and novelty of this paper is not very clear. Due to the following reasons, I cannot recommend this manuscript for publication on the Royal Society Open Science:

1. There is no clear problem formulation and analysis in the manuscript. I guess that this manuscript just review some issues on Gaussian processes and stochastic problems.
- 2- The structure of the introduction and whole manuscript are disordered and needs to be reorganized.
- 3- The results and simulations are not clear to verify the derived achievements. The results must be structured in the form of theorems, remarks,

Review form: Reviewer 2

Is the manuscript scientifically sound in its present form?

Yes

Are the interpretations and conclusions justified by the results?

No

Is the language acceptable?

Yes

Do you have any ethical concerns with this paper?

No

Have you any concerns about statistical analyses in this paper?

No

Recommendation?

Major revision is needed (please make suggestions in comments)

Comments to the Author(s)

First, a necessary disclosure. I am a power engineer, not a mathematician, so my interest is from the application point of view. Hence, I am not able to assess the presented mathematics and my assessment is concentrated on whether or not the presented methodology could be of interest from power system application point of view. Generally, the answer to that question is yes (providing that the methodology works, about which later). The author claims two main advantages of the presented methodology. Firstly, it utilises Kalman filter to process information as it comes along. Secondly, it processes information from many measurement points (Phasor Measurement Units, PMUs) simultaneously. This compares favourably with currently used methodologies which typically operate on a moving time window (i.e. operate in a batch mode) and process information from one PMU at a time. The latter means that it may be difficult to identify the same mode at different locations as its estimated frequency and damping may have slightly different values in different locations.

The main comment about the paper is that the methodology has not been properly validated, i.e. we do not really know whether it works or not. For testing his method, the author used data presented in Fig. 4. There are two problems with that data. The first is that Fig. 4 presents a frequency trace with 1 Hz time resolution while PMU measurements are taken with the frequency of about 30-60 Hz. Resolution of 1 Hz is not high enough for the estimation of power system dynamics. Secondly, the data is from a single measurement point so it is not possible to check the author's claimed benefit of being able to process measurements from many PMUs simultaneously.

The author claims that he could not use National Grid data due to confidentiality reasons but it is unclear why he did not use Fig. 1 which presumably shows actual PMU data and has a much higher time resolution than Fig. 4. Also, Fig. 4 shows data from 8 PMUs so the author could check his claim that his method could process data from many PMUs simultaneously.

Generally, the problem of not being able to access real data is quite common and a standard approach is then to simulate it. This is quite simple and would require simulating a simple power system using standard software (e.g. Matpower, PSAT, PST or any other), add noise to the signals, estimate the dominant modes and compare them to the actual ones (known as we know the underlying model). The advantage of using simulated data is that we know the actual model so we can check how accurate the estimates are.

The second comment is editorial. The paper presents a general, textbook-like treatment of the problem rather than a derivation of a methodology. The treatment is quite wide (46 pages), aiming to be complete, with a sprawling narrative and many diversions. The reader is distracted from the main message and has to go through a lot of supplementary material which is often only marginally related to, or used for, the proposed methodology (for example, sections 3 and 4 - Langevin process, OU process). I am not sure if such a comprehensive treatment is acceptable (or perhaps even desired) for this journal but I personally would prefer a more disciplined and focused treatment concentrating on the derivation of the methodology and relegating a lot of background and supplementary material to appendices.

To summarise, the paper presents an interesting and potentially useful piece of mathematics but its practical usefulness is still to be proved. In my personal opinion, the presentation could be made more focused but I would defer to the Editor and the author to decide whether or not to follow this recommendation.

Review form: Reviewer 3

Is the manuscript scientifically sound in its present form?

Yes

Are the interpretations and conclusions justified by the results?

Yes

Is the language acceptable?

Yes

Do you have any ethical concerns with this paper?

No

Have you any concerns about statistical analyses in this paper?

No

Recommendation?

Accept as is

Comments to the Author(s)

Thanks, I enjoyed your introductory review and the application to a power grid and possible other applications.

Review form: Reviewer 4

Is the manuscript scientifically sound in its present form?

Yes

Are the interpretations and conclusions justified by the results?

Yes

Is the language acceptable?

Yes

Do you have any ethical concerns with this paper?

No

Have you any concerns about statistical analyses in this paper?

No

Recommendation?

Accept with minor revision (please list in comments)

Comments to the Author(s)

Motivated by National Grid's need to detect nascent oscillations in power flow fast enough to activate controller strategies, the author considers and develops a method of detecting oscillations in a fluctuating signal in real time. He does this through treatment of a model dynamical system subject to Gaussian noise.

I found it difficult to get an intuitive feeling for the extent to which the approach has succeeded in solving the original problem. For example, looking at Fig 1, the oscillations become evident by eye at about 608 or 609 s, though a little earlier on one of the green traces. So how much sooner than this can the new method reliably identify growth of the incipient oscillations?

This is an interesting, scholarly, and highly erudite manuscript. So far as I can tell, it is correct and original and meets all of the RSOS criteria, so I am happy to recommend that it be accepted for publication.

A couple of typos:

Page/line	Comment
3/17	"processes"
32/34	"data are" [data are plural - there are also other instances]

Review form: Reviewer 5

Is the manuscript scientifically sound in its present form?

Yes

Are the interpretations and conclusions justified by the results?

No

Is the language acceptable?

No

Do you have any ethical concerns with this paper?

No

Have you any concerns about statistical analyses in this paper?

No

Recommendation?

Major revision is needed (please make suggestions in comments)

Comments to the Author(s)

The author has written an interesting paper on a technique that could have potential within power systems. The paper is quite long with much general information before we come to the core method. The background information is useful but such a relatively detailed treatment could have been moved to an appendix. The paper is written in a bit personal style and I presume that this is acceptable for a uni-personal paper but possible there should be no reference to family members and co-workers. The paper will benefit from a small example demonstrating the approach. I would like to compliment the author for clear statements of the deficiencies of the current implementation. Maybe statements like this need to be removed: (we might extend this unpublished paper to include a report on some tests).

Decision letter (RSOS-201442.R0)

Dear Dr MacKay

The Editors assigned to your paper RSOS-201442 "Inference of dominant modes for linear stochastic processes" have now received comments from reviewers and would like you to revise the paper in accordance with the reviewer comments and any comments from the Editors. Please note this decision does not guarantee eventual acceptance.

We invite you to respond to the comments supplied below and revise your manuscript. Below the referees' and Editors' comments (where applicable) we provide additional requirements. Final acceptance of your manuscript is dependent on these requirements being met. We provide guidance below to help you prepare your revision. While tweaking the paper into journal style is

helpful, this is not strictly necessary at this stage in the process - in the event the paper is accepted for publication, the production team will be able to assist with this. Similarly, amending the reference list may aid clarity at this stage, but as long as a DOI is included (where applicable), our production office can put the bibliography into an appropriate format in due course.

Please submit your revised manuscript and required files (see below) no later than 21 days from today's (ie 18-Jan-2021) date. Note: the ScholarOne system will 'lock' if submission of the revision is attempted 21 or more days after the deadline. If you do not think you will be able to meet this deadline please contact the editorial office immediately.

on behalf of Dr Derek Abbott (Associate Editor) and Mark Chaplain (Subject Editor)
openscience@royalsociety.org

Associate Editor Comments to Author (Dr Derek Abbott):

Associate Editor: 1

Comments to the Author:

- a) Please use the journal's LaTeX template
- b) Please use the journal's referencing style
- c) Please remove the outer bounding boxes from Figs 4 and 5.
- d) Figs 4 and 5 appear to be from a lower quality graphing package than your other graphs. Please use the better graphing package throughout for uniform quality of presentation.

Associate Editor: 2

Comments to the Author:

This paper proposes a new theory to search inference of dominant modes for linear stochastic process. The argument, however, is not sufficient to make the reader to be convinced, of its advantage and novelty, both in comparison with other theories and examples.

Reviewer comments to Author:

Reviewer: 1

Comments to the Author(s)

This paper addresses the development of methods to infer dynamical systems dominant modes from observations in real time. Overall, the contribution and novelty of this paper is not very

clear. Due to the following reasons, I cannot recommend this manuscript for publication on the Royal Society Open Science:

1. There is no clear problem formulation and analysis in the manuscript. I guess that this manuscript just review some issues on Gaussian processes and stochastic problems.
- 2- The structure of the introduction and whole manuscript are disordered and needs to be reorganized.
- 3- The results and simulations are not clear to verify the derived achievements. The results must be structured in the form of theorems, remarks,

Reviewer: 2

Comments to the Author(s)

First, a necessary disclosure. I am a power engineer, not a mathematician, so my interest is from the application point of view. Hence, I am not able to assess the presented mathematics and my assessment is concentrated on whether or not the presented methodology could be of interest from power system application point of view. Generally, the answer to that question is yes (providing that the methodology works, about which later). The author claims two main advantages of the presented methodology. Firstly, it utilises Kalman filter to process information as it comes along. Secondly, it processes information from many measurement points (Phasor Measurement Units, PMUs) simultaneously. This compares favourably with currently used methodologies which typically operate on a moving time window (i.e. operate in a batch mode) and process information from one PMU at a time. The latter means that it may be difficult to identify the same mode at different locations as its estimated frequency and damping may have slightly different values in different locations.

The main comment about the paper is that the methodology has not been properly validated, i.e. we do not really know whether it works or not. For testing his method, the author used data presented in Fig. 4. There are two problems with that data. The first is that Fig. 4 presents a frequency trace with 1 Hz time resolution while PMU measurements are taken with the frequency of about 30-60 Hz. Resolution of 1 Hz is not high enough for the estimation of power system dynamics. Secondly, the data is from a single measurement point so it is not possible to check the author's claimed benefit of being able to process measurements from many PMUs simultaneously.

The author claims that he could not use National Grid data due to confidentiality reasons but it is unclear why he did not use Fig. 1 which presumably shows actual PMU data and has a much higher time resolution than Fig. 4. Also, Fig. 4 shows data from 8 PMUs so the author could check his claim that his method could process data from many PMUs simultaneously.

Generally, the problem of not being able to access real data is quite common and a standard approach is then to simulate it. This is quite simple and would require simulating a simple power system using standard software (e.g. Matpower, PSAT, PST or any other), add noise to the signals, estimate the dominant modes and compare them to the actual ones (known as we know the underlying model). The advantage of using simulated data is that we know the actual model so we can check how accurate the estimates are.

The second comment is editorial. The paper presents a general, textbook-like treatment of the problem rather than a derivation of a methodology. The treatment is quite wide (46 pages), aiming to be complete, with a sprawling narrative and many diversions. The reader is distracted from the main message and has to go through a lot of supplementary material which is often only

marginally related to, or used for, the proposed methodology (for example, sections 3 and 4 - Langevin process, OU process). I am not sure if such a comprehensive treatment is acceptable (or perhaps even desired) for this journal but I personally would prefer a more disciplined and focused treatment concentrating on the derivation of the methodology and relegating a lot of background and supplementary material to appendices.

To summarise, the paper presents an interesting and potentially useful piece of mathematics but its practical usefulness is still to be proved. In my personal opinion, the presentation could be made more focused but I would defer to the Editor and the author to decide whether or not to follow this recommendation.

Reviewer: 3

Comments to the Author(s)

Thanks, I enjoyed your introductory review and the application to a power grid and possible other applications.

Reviewer: 4

Comments to the Author(s)

Motivated by National Grid's need to detect nascent oscillations in power flow fast enough to activate controller strategies, the author considers and develops a method of detecting oscillations in a fluctuating signal in real time. He does this through treatment of a model dynamical system subject to Gaussian noise.

I found it difficult to get an intuitive feeling for the extent to which the approach has succeeded in solving the original problem. For example, looking at Fig 1, the oscillations become evident by eye at about 608 or 609 s, though a little earlier on one of the green traces. So how much sooner than this can the new method reliably identify growth of the incipient oscillations?

This is an interesting, scholarly, and highly erudite manuscript. So far as I can tell, it is correct and original and meets all of the RSOS criteria, so I am happy to recommend that it be accepted for publication.

A couple of typos:

Page/line	Comment
3/17	"processes"
32/34	"data are" [data are plural - there are also other instances]

Reviewer: 5

Comments to the Author(s)

The author has written an interesting paper on a technique that could have potential within power systems. The paper is quite long with much general information before we come to the core method. The background information is useful but such a relatively detailed treatment could have been moved to an appendix. The paper is written in a bit personal style and I presume that this is acceptable for a uni-personal paper but possible there should be no reference to family members and co-workers. The paper will benefit from a small example demonstrating the approach. I would like to compliment the author for clear statements of the deficiencies of the current implementation. Maybe statements like this need to be removed: (we might extend this unpublished paper to include a report on some tests).

===PREPARING YOUR MANUSCRIPT===

===PREPARING YOUR REVISION IN SCHOLARONE===

- An individual file of each figure (EPS or print-quality PDF preferred [either format should be produced directly from original creation package], or original software format).
 - An editable file of each table (.doc, .docx, .xls, .xlsx, or .csv).
 - An editable file of all figure and table captions.
- Note: you may upload the figure, table, and caption files in a single Zip folder.
- Any electronic supplementary material (ESM).
 - If you are requesting a discretionary waiver for the article processing charge, the waiver form must be included at this step.
 - If you are providing image files for potential cover images, please upload these at this step, and inform the editorial office you have done so. You must hold the copyright to any image provided.
 - A copy of your point-by-point response to referees and Editors. This will expedite the preparation of your proof.

- Ensure that your data access statement meets the requirements at <https://royalsociety.org/journals/authors/author-guidelines/#data>. You should ensure that you cite the dataset in your reference list. If you have deposited data etc in the Dryad repository, please include both the 'For publication' link and 'For review' link at this stage.
- If you are requesting an article processing charge waiver, you must select the relevant waiver option (if requesting a discretionary waiver, the form should have been uploaded at Step 3 'File upload' above).
- If you have uploaded ESM files, please ensure you follow the guidance at <https://royalsociety.org/journals/authors/author-guidelines/#supplementary-material> to include a suitable title and informative caption. An example of appropriate titling and captioning may be found at https://figshare.com/articles/Table_S2_from_Is_there_a_trade-off_between_peak_performance_and_performance_breadth_across_temperatures_for_aerobic_sc_ope_in_teleost_fishes_/3843624.

Author's Response to Decision Letter for (RSOS-201442.R0)

See Appendix A.

Decision letter (RSOS-201442.R1)

Dear Dr MacKay,

It is a pleasure to accept your manuscript entitled "Inference of dominant modes for linear stochastic processes" in its current form for publication in Royal Society Open Science.

on behalf of Dr Derek Abbott (Associate Editor) and Mark Chaplain (Subject Editor)
openscience@royalsociety.org

Appendix A

RSOS-201442 Inference of dominant modes for linear stochastic systems

I am most grateful for the reviewer reports. I have made a thorough rewrite, moving the pedagogical material to appendices and streamlining the main presentation.

Associate Editor Comments to Author (Dr Derek Abbott):

Associate Editor: 1

Comments to the Author:

a) Please use the journal's LaTeX template: *done*

b) Please use the journal's referencing style: *done*

c) Please remove the outer bounding boxes from Figs 4 and 5. *done*

d) Figs 4 and 5 appear to be from a lower quality graphing package than your other graphs. Please use the better graphing package throughout for uniform quality of presentation.
done

Associate Editor: 2

Comments to the Author:

This paper proposes a new theory to search inference of dominant modes for linear stochastic process. The argument, however, is not sufficient to make the reader to be convinced, of its advantage and novelty, both in comparison with other theories and examples.

I have enlarged the discussion of comparison with other methods. It may still not be convincing but at least gives the reader a wider view of what alternatives there are.

Reviewer comments to Author:

Reviewer: 1

Comments to the Author(s)

This paper addresses the development of methods to infer dynamical systems dominant modes from observations in real time. Overall, the contribution and novelty of this paper is not very clear.

I've reorganised the paper to make the contribution and novelty clearer.

Due to the following reasons, I cannot recommend this manuscript for publication on the Royal Society Open Science:

1. There is no clear problem formulation and analysis in the manuscript. I guess that this manuscript just review some issues on Gaussian processes and stochastic problems.

I believe I have given a clear problem formulation and analysis, but I've moved the reviews to appendices to make the main problem and solution stand out better.

2- The structure of the introduction and whole manuscript are disordered and needs to be reorganized.

I've totally reorganised the structure.

3- The results and simulations are not clear to verify the derived achievements.

The paper does not include any simulations. Testing of the proposed method is a future project.

The results must be structured in the form of theorems, remarks,

I feel this is not appropriate for this paper. It would give a misleading impression that it is all about rigorous formulation and proofs. The results are variants on known results and are

methods, so hardly merit calling theorems. It is more about a philosophy: to use linear stochastic processes rather than more general Gaussian processes, and to fit to a mode process rather than a system.

Reviewer: 2

Comments to the Author(s)

First, a necessary disclosure. I am a power engineer, not a mathematician, so my interest is from the application point of view. Hence, I am not able to assess the presented mathematics and my assessment is concentrated on whether or not the presented methodology could be of interest from power system application point of view. Generally, the answer to that question is yes (providing that the methodology works, about which later). The author claims two main advantages of the presented methodology. Firstly, it utilises Kalman filter to process information as it comes along. Secondly, it processes information from many measurement points (Phasor Measurement Units, PMUs) simultaneously. This compares favourably with currently used methodologies which typically operate on a moving time window (i.e. operate in a batch mode) and process information from one PMU at a time. The latter means that it may be difficult to identify the same mode at different locations as its estimated frequency and damping may have slightly different values in different locations.

I am pleased that the reviewer appreciates these two points, which are indeed two of my main points.

The main comment about the paper is that the methodology has not been properly validated, i.e. we do not really know whether it works or not.

I agree and am clear about this. I feel the method is worth publishing, rather than waiting for me to find a student willing to try it out, so that someone else could try it. National Grid felt the problem was quite urgent and it is already 6 years since they posed it.

For testing his method, the author used data presented in Fig. 4.

I would not call that a test of the method. It was to get a feel for a suitable assumption about the power imbalance noise.

There are two problems with that data. The first is that Fig. 4 presents a frequency trace with 1 Hz time resolution while PMU measurements are taken with the frequency of about 30-60 Hz. Resolution of 1 Hz is not high enough for the estimation of power system dynamics.

I agree and made that point in the paper, but National Grid chose to make the data publicly available at only 1 second resolution.

Secondly, the data is from a single measurement point so it is not possible to check the author's claimed benefit of being able to process measurements from many PMUs simultaneously.

I agree. National Grid were going to give me access to simultaneous data but terminated my contract.

The author claims that he could not use National Grid data due to confidentiality reasons but it is unclear why he did not use Fig. 1 which presumably shows actual PMU data and has a much higher time resolution than Fig. 4. Also, Fig. 4 shows data from 8 PMUs so the

author could check his claim that his method could process data from many PMUs simultaneously.

I do not have the data for Fig.1. It is just copied from the cited paper.

Generally, the problem of not being able to access real data is quite common and a standard approach is then to simulate it. This is quite simple and would require simulating a simple power system using standard software (e.g. Matpower, PSAT, PST or any other), add noise to the signals, estimate the dominant modes and compare them to the actual ones (known as we know the underlying model). The advantage of using simulated data is that we know the actual model so we can check how accurate the estimates are.

Yes, it would be a good idea to test the method on simulated data. Indeed, I proposed at the end of the AC power section that the next step would be to implement the method for a two-node system and it would be sensible to test this on simulated data.

The second comment is editorial. The paper presents a general, textbook-like treatment of the problem rather than a derivation of a methodology.

It does derive a methodology, but I agree that it is perhaps hidden by too much pedagogical material.

The treatment is quite wide (46 pages),

Only 28 in RSOS style

aiming to be complete, with a sprawling narrative and many diversions. The reader is distracted from the main message and has to go through a lot of supplementary material which is often only marginally related to, or used for, the proposed methodology (for example, sections 3 and 4 - Langevin process, OU process). I am not sure if such a comprehensive treatment is acceptable (or perhaps even desired) for this journal but I personally would prefer a more disciplined and focused treatment concentrating on the derivation of the methodology and relegating a lot of background and supplementary material to appendices.

I have reorganised the material to make the main text streamlined to highlight the main points and moved the pedagogical material to appendices.

To summarise, the paper presents an interesting and potentially useful piece of mathematics but its practical usefulness is still to be proved.

A fair summary!

In my personal opinion, the presentation could be made more focused but I would defer to the Editor and the author to decide whether or not to follow this recommendation.

I have taken your recommendation and focussed the presentation.

Reviewer: 3

Comments to the Author(s)

Thanks, I enjoyed your introductory review and the application to a power grid and possible other applications.

Reviewer: 4

Comments to the Author(s)

Motivated by National Grid's need to detect nascent oscillations in power flow fast enough to activate controller strategies, the author considers and develops a method of detecting oscillations in a fluctuating signal in real time. He does this through treatment of a model dynamical system subject to Gaussian noise.

I found it difficult to get an intuitive feeling for the extent to which the approach has succeeded in solving the original problem. For example, looking at Fig 1, the oscillations become evident by eye at about 608 or 609 s, though a little earlier on one of the green traces. So how much sooner than this can the new method reliably identify growth of the incipient oscillations?

This is a good question. I don't know. If I set the time-constant for the memory of parameters too long then I won't be able to detect changes on a shorter timescale, so it may be better to adopt a model for the parameter variation that allows jumps. That is something for the future.

This is an interesting, scholarly, and highly erudite manuscript. So far as I can tell, it is correct and original and meets all of the RSOS criteria, so I am happy to recommend that it be accepted for publication.

A couple of typos:

Page/line	Comment
3/17	"processes"
32/34	"data are" [data are plural – there are also other instances] done

Reviewer: 5

Comments to the Author(s)

The author has written an interesting paper on a technique that could have potential within power systems. The paper is quite long with much general information before we come to the core method. The background information is useful but such a relatively detailed treatment could have been moved to an appendix.

Done

The paper is written in a bit personal style and I presume that this is acceptable for a uni-personal paper but possible there should be no reference to family members and co-workers.

I prefer to keep the references to my brother.

The paper will benefit from a small example demonstrating the approach.

I agree, but that requires implementing the method and I prefer to find someone else to do that.

I would like to compliment the author for clear statements of the deficiencies of the current implementation.

Maybe statements like this need to be removed: (we might extend this unpublished paper to include a report on some tests).

Done

===PREPARING YOUR MANUSCRIPT===

Mostly done.

Not done because I have so extensively reorganised the manuscript.

Done

Please ensure that any equations included in the paper are editable text and not embedded images. *Yes*

Please ensure that you include an acknowledgements' section before your reference list/bibliography. This should acknowledge anyone who assisted with your work, but does not qualify as an author per the guidelines at <https://royalsociety.org/journals/ethics-policies/openness/>. *Yes*

While not essential, it will speed up the preparation of your manuscript proof if accepted if you format your references/bibliography in Vancouver style (please see <https://royalsociety.org/journals/authors/author-guidelines/#formatting>).

Mostly done. Just I didn't put the year where you'd like it.

You should include DOIs for as many of the references as possible.

I've done enough. I consider such things to be up to the journal
